# Sleep-Body Composition Relationship: Roles of Sleep Behaviors in General and Abdominal Obesity in Chinese Adolescents Aged 17–22 Years

**DOI:** 10.3390/nu15194130

**Published:** 2023-09-25

**Authors:** Yalin Song, Lu Gong, Xiaomin Lou, Huijun Zhou, Yudan Hao, Qiuyuan Chen, Yize Zhao, Xili Jiang, Lijie Li, Xian Wang

**Affiliations:** College of Public Health, Zhengzhou University, No. 100 Science Avenue, Zhengzhou 450001, China; songyalin@zzu.edu.cn (Y.S.); gl17355057205@163.com (L.G.); louxm@zzu.edu.cn (X.L.); m18232128356@163.com (H.Z.); haoyudan_921@163.com (Y.H.); chenqiuyuan_72@foxmail.com (Q.C.); 202052010439@stu.zzu.edu.cn (Y.Z.); jiangxili1@163.com (X.J.); 13395925785@163.com (L.L.)

**Keywords:** obesity, body composition, nap duration, social jetlag, screen time before sleep

## Abstract

This study aimed to investigate the association between sleep behaviors and body composition, which was measured by bioelectrical impedance analysis (BIA) among Chinese adolescents. Overall, 444 students (65.3% females, 19.12 ± 1.177 years) completed questionnaires describing sleep characteristics. Sleep characteristics were derived from subjective means. Body composition was obtained from BIA by InBody 720 (Biospace Co. Ltd., Seoul, Republic of Korea). Regression models tested relationships between sleep and body composition after adjustment for covariates. Students with weekday nap duration (>30 min/d) exerted higher waist-height ratio (WHtR) (*B* = 0.013, FDR-corrected *p* = 0.080). Average sleep duration (≤7 h/d) was linked to more WHtR (*B* = 0.016, FDR-corrected *p* = 0.080). People with high social jetlag showed gained visceral fat area (*B* = 7.475), WHtR (*B* = 0.015), waist to hip ratio (*B* = 0.012), fat mass index (*B* = 0.663) and body fat percentage (*B* = 1.703) (all FDR-corrected *p* < 0.1). Individuals with screen time before sleep (>0.5 h) exhibited higher visceral fat area (*B* = 7.934, FDR-corrected *p* = 0.064), WHtR (*B* = 0.017, FDR-corrected *p* = 0.080), waist to hip ratio (*B* = 0.016, FDR-corrected *p* = 0.090), fat mass index (*B* = 0.902, FDR-corrected *p* = 0.069) and body fat percentage (*B* = 2.892, FDR-corrected *p* = 0.018). We found poor sleep characteristics were closely related to general and abdominal obesity.

## 1. Introduction

The prevalence of overweight and obesity has been increasingly serious during the last four decades, which has undoubtedly become a major public health concern in China. The national survey in 2018 demonstrated that the obesity rate of Chinese adolescents reached 16.0% [1,2]. Abdominal obesity is defined as excess fat distribution in the abdominal area [3]. Waist-to-height ratio (WHtR), a maker of central adiposity, has strong associations with diabetes, hypertension [4] and cardiometabolic health [5]. A study involving college students recruited from a campus in Beijing, China, underlined the necessity of fat mass (FM) and fat free mass (FFM) control to avoid the risk of premature death. Abundant FM and low FFM represent general fat accumulation, which has adverse health effects especially in adolescents [6]. Compared to normal-weight peers, adolescents classified as obese have higher rates of morbidity and mortality, and higher risks of developing obesity as adults [1]. More attention should be paid to the population in establishing public health strategies. The novel coronavirus disease 2019 (COVID-19) has spread rapidly throughout countries. Available evidence suggested that adolescents with obesity were more inclined to develop severe medical conditions from COVID-19 [7,8]. Therefore, to confront the COVID-19 pandemic, it is of great significance to focus on general and abdominal obesity in adolescents.

Body composition is considered as an important predictor in various clinical scenarios including general and abdominal obesity. Body mass index (BMI) is a widely used indicator for defining obesity [9,10]. Based on Chinese criteria, overweight is defined as a BMI of 24.0 kg/m^2^ and obesity as a BMI of 28.0 kg/m^2^ or higher in adults (≥18 years) [11,12]. Nonetheless, BMI does not take into account the distribution of adipose tissue, and cannot distinguish between FM and FFM [9], which in several situations is the key factor influencing disease risk. For instance, visceral adipose tissue is known to be more closely linked to cardiovascular risk than subcutaneous adipose tissue, thus it is more beneficial to assess waist circumference (WC) or waist to hip ratio (WHR) instead of BMI [13]. WC, WHtR and WHR are key markers for identifying abdominal obesity [14]. Bioelectrical impedance analysis (BIA) and dual energy X-ray absorptiometry (DXA) are extensively used settings to evaluate body composition mainly in epidemiological and clinical analysis, respectively. DXA has better accuracy compared to BIA, but the safety of repeated measurements is worth improving [15].

The causes for excessive fat accumulation and high rise of obesity are complicated with biological, economic, social and cultural factors, and negative effects of poor sleep may be part of the determinants [16]. We spend a third of our day sleeping as it plays an essential part in our life [17]. A population-based longitudinal study with 1024 participants has shown that insufficient sleep leads to reduced leptin and elevated ghrelin levels, which can increase appetite, thus promoting weight gain [18]. Of note, the COVID-19 pandemic has exacerbated the prevalence of sleep problems, which may contribute to shorter sleep duration, poor sleep quality, low sleep efficiency, etc., in all populations, especially the groups of children and adolescents [19,20]. Insufficient sleep duration and low sleep efficiency were associated with digestive disorders or conversely represented specific clinical manifestations of gastrointestinal diseases [21]. Studies conducted in China, Spain, the USA, and the UK demonstrated a significant increase in obesity occurrence when nap duration lasted longer than an hour [22]. It seems that individual’s daytime napping is not always constant and may vary between weekdays and weekends. An early school start time and a heavy study burden may contribute to a student’s fatigue so that a short break is needed.

Social jetlag (SJL) is a dissonance between the biological clock and the social clock, which stems from students’ desire to make up for sleep loss during the weekdays by waking up later on weekends [23], especially in E-types, who have a later bedtime/wake-up time, and reach their best achievements during the second half of the day [24]. A study of adolescents (45% females, 9–17 years) emphasized SJL was related to obvious changes in fat mass index (FMI) [25]. However, another study containing 390 healthy adults (21–35 years) exhibited that SJL had no influence on any obesity-related anthropometric indices [26].

Owing to growing autonomy or changes in learning methods, college students tend to have an increase in screen media exposure [27], consequently with a high risk of obesity [28]. A cross-sectional study with 49,051 college and university students showed that screen use in bed had a strong negative association with sleep latency, sleep duration and sleep efficiency [29]. Therefore, due to adolescents’ preferred and unique pattern of biological rhythm, it is of great significance to understand how poor sleep characteristics yield impacts on obesity and body composition for the purpose of taking preventive measures and designing appreciated strategies.

Overall, previous studies have supported a role of poor sleep in the development of obesity [30,31,32]. Literatures regarding sleep duration [33] or obstructive sleep apnea [34,35] merely used BMI and WC to define general and abdominal obesity. A majority of researchers have limited parameters in identifying general and central obesity. The present study took FMI, fat free mass index (FFMI), BMI, BFP, WHR and WHtR into account using Inbody 720, an analyzer utilizing a tetrapolar 8-point tactile electrode system and hand-to-foot BIA that sends varying frequencies of alternating current through the body, to make an accurate and generalized estimation for obesity. As adolescents are in a period of rapid change [36], it is necessary to focus on the unique population. Of note, there is a gap in the research about roles of poor sleep in the body composition in adolescents (aged 17–22 years), especially in China, the world’s most populous nation.

The present study aimed to evaluate the relationship between sleep characteristics and body composition in Chinese adolescents aged 17–22 years. Not only did we observe the various novel sleep variables but we also paid attention to the weekday-to-weekend sleep differences such as SJL in this special population. Additionally, we examined sex difference in sleep characteristics together with body composition parameters and adjusted for gender in linear regression analyses. Notably, the definition of adolescence was prolonged in the current study.

## 2. Materials and Methods

### 2.1. Participants

A total of 469 healthy students from freshman to senior, aged 17–22 years, from the School of Pharmacy, School of Public Health and School of Stomatology of Zhengzhou University (Zhengzhou, China) were recruited using a convenience sample in this cross-sectional observational study. Exclusion criteria were contraindications to BIA, such as diseases affecting the electrical resistance of the skin, pregnancy, having an implanted pacemaker, and the presence of metal prostheses. Individuals with endocrine or renal diseases, cancer or sever inflammatory conditions were not included in the sample. In the study, due to the application of BIA, all students had exact body composition data. Participants who did not have complete important data were excluded (*n* = 22). We excluded 3 adolescents whose midpoints of sleep during weekdays were between 22:00 and 24:00 because their data were invalid. Overall, 25 participants were excluded from the analysis, and 444 students met the criteria. The study protocol was approved by the Life Science Ethics Committee of Zhengzhou University (ZZUIRB2021-94) and informed consent was obtained from all participants.

### 2.2. Study Protocol

Sleep variables, demographic characteristics and living habit characteristics were derived from subjective means. A questionnaire survey was administered to participants under the supervision of professional investigators. Additionally, anthropometric and bioelectrical measurements were carried out at about 7:00 p.m. every day at an appropriate space from 8 November 2021 to 21 November 2021. Height was measured without shoes to the nearest 0.1 cm using a stadiometer fixed to the wall. WC was measured using a non-elastic tape at the midpoint line between the lowest point of the rib and the upper edge of the iliac crest, to the nearest 0.1 cm. Body composition parameters were precisely measured by BIA using the InBody 720 (Biospace Co., Ltd., Seoul, Republic of Korea). Before measurements were taken, participants were asked to void their bladders, remove any jewelry, take off shoes and socks, stand in a designated area of the instrument and place the forefeet on the front sole electrodes and the heels on the rear sole electrodes. During the measurement, participants placed their palms on the palm electrodes and pressed their thumbs on the thumb electrodes to contact the current. Additionally, in order to obtain reliable test data, it was necessary to keep the room temperature constant.

### 2.3. Sleep Characteristics

Weekday and weekend nap duration were obtained from “In the last month, how many minutes did you nap on average ?” with response options of ≤30 min/d or >30 min/d. Subjects were also asked to recall the questions (in the last month), “What time did you go to bed on average at night on weekdays and weekends?”; “How many minutes on average did you fall asleep after going to bed on weekdays and weekends?” and “What time did you get up on average in the morning on weekdays and weekends?”. Average sleep latency was calculated as (weekday sleep latency × 5 + weekend sleep latency × 2)/7. Sleep duration was defined as average time to wake up—average time to bed—average sleep latency and was divided into ≤7 h/d and >7 h/d [37]. Similarly, average sleep duration was calculated as (weekday sleep duration × 5 + weekend sleep duration × 2)/7. Average sleep efficiency was defined as a ratio of total sleep time to time in bed (×100%) [38] and a higher percentage indicated better sleep quality. Screen time before sleep (>0.5 h) (no vs. yes) was estimated by the following question: “Was your screen time before sleep more than 0.5 h at night in the last month?”. SJL was defined as the difference in hours between the midpoints of sleep on weekdays (school days) versus weekends (free days) [23]. This was calculated by subtracting the midpoint of sleep on weekdays from that of weekends [39] and was divided into ≤1 h and >1 h [40].

### 2.4. Body Composition Assessment

Height and WC were calculated by a stadiometer or a non-elastic tape. Weight, hip circumference (HC), FM, FFM, visceral fat area (VFA), BMI, WHtR, WHR and BFP were assessed by BIA using Inbody 720. BMI was defined as weight (kg) divided by squared height (m^2^). WHtR was calculated as WC/height and WHR was calculated as WC/HC. FMI was calculated as FM/square of the height (kg/m^2^) and FFMI was calculated as FFM/square of the height (kg/m^2^). BFP was defined as the proportion of fat weight in total weight. All anthropometric measurements were conducted under the guidance of trained research assistants.

### 2.5. Covariates

Information on demographic characteristics was collected through the self-administered questionnaire: gender, father’s education (below elementary school, elementary school, junior school, high school or technical secondary school, junior college or above), mother’s education (below elementary school, elementary school, junior school, high school or technical secondary school, junior college or above), race (Han vs. minority), location (rural vs. urban), the only child (no vs. yes), self-rated family income (low, middle or high).

Likewise, living habit characteristics were assessed in a subjective way. For instance, “takeaway food consumption/week” was estimated from “In the last week, how many servings did you have takeaway food on average?” with response options of: “none”, “1–2”, “2–4” and “>4”. Subsequently, we merged the last two options as >2. Breakfast consumption (<7 days/week vs. 7 days/week), vegetables consumption/d (<3 servings vs. ≥3 servings), fruits consumption/d (<1 servings, 1 serving or >1 servings), dried fruits consumption/d (0, <1 servings or ≥1 servings), pure juice consumption (>250 mL)/d (0, <1 servings or ≥1 servings) and smoking use (no vs. yes) in the last week were assessed. Similarly, in the last month, soft drinks consumption (>250 mL)/week (0, 1 serving and >1 servings), sugar-sweetened beverage consumption (>250 mL)/week (0, 1 serving and >1 servings), alcohol consumption (no vs. yes), duration of physical exercise each time (<60 min vs. ≥60 min), number of physical exercise (<1/d vs. ≥1/d), weekday screen time/d (<2 h, <4 h or ≥4 h) and weekend screen time/d (<2 h, <4 h or ≥4 h) were assessed.

### 2.6. Statistical Analysis

Data were excluded from all analyses if missing any of the variables. Categorical variables including demographic characteristics, living habit characteristics and sleep variables were expressed as number (%) using Chi-square tests or Fisher’s exact tests stratified by sex to test. All continuous data including several sleep variables and body composition parameters were examined for normality by the Shapiro–Wilk method, reported as mean and standard deviation. Sex differences between body composition indices were analyzed with unpaired *t*-test. For the purpose of determining confounding factors, unpaired *t*-test and univariate one-way ANOVAs were conducted between the dependent variables and possible confounders (demographic characteristics and living habits characteristics).

In order to investigate relevance between sleep characteristics and changes in body composition indices, linear regression analyses including a primary model (model 1) and a multivariable model (model 2) were performed, whose outcomes were reported as unstandardized coefficients (*B*). In univariable models, sleep variables comprising weekday nap duration/d, weekend nap duration/d, average sleep duration/d, etc., were the independent predictors. Each sleep variable was initially considered separately. In adjusted models, the following confounders potentially affecting the associations were considered: gender, father’s education, mother’s education, breakfast consumption, fruits consumption, pure juice consumption, soft drinks consumption, alcohol consumption, smoking and duration of physical exercise each time. Point and interval estimates of statistical significance were exhibited as *p*-value/*P* and 95% *CI*, with *p* < 0.05 considered significant. All statistical analyses were performed using IBM SPSS Statistics software version 21.0. The Benjamini–Hochberg method was used to control the false discovery rate (FDR) for the multiple linear regression analyses with FDR-corrected *p* < 0.1 considered significant [41].

## 3. Results

In our study, 444 participants (154 males and 290 females) were enrolled. The mean age was 19.12 ± 1.177 years old.

### 3.1. General Characteristics Classified by Gender

As presented in Table 1 and Table 2, demographic characteristics (race, location, the only child, father’s education, mother’s education, self-rated family income) did not differ between male and female participants. Among living habit characteristics, males consumed a lower quantity of breakfast (*χ*^2^ = 5.369) and fruits (*χ*^2^ = 16.067), and a higher quantity of takeaway food (*χ*^2^ = 14.944), pure juice (*χ*^2^ = 11.307), soft drinks (>250 mL) (*χ*^2^ = 15.286) and alcohol (*χ*^2^ = 26.115) than females (all *p* < 0.05). There was a significant reduction in the frequency of smoking (*χ*^2^ = 7.798, *p* = 0.005) and duration of physical exercise each time (*χ*^2^ = 13.190, *p* < 0.001) in females compared with males. Both males and females had the highest frequency in weekend screen time/d (≥4 h) (*χ*^2^ = 15.653, *p* = 0.016). Participants stratified by sex showed no differences in other living habit characteristics.

### 3.2. Sleep Characteristics Including Social Jetlag, etc. Classified by Gender

As illustrated in Table 3, males tended to have longer weekday nap duration (>30 min) (*χ*^2^ = 13.852, *p* < 0.001) and SJL (>1 h) (*χ*^2^ = 5.846, *p* = 0.016) in comparison with females. There were no differences in weekend nap duration, average sleep duration, average sleep efficiency and screen time before sleep (>0.5 h) between men and women.

### 3.3. Body Composition Parameters Classified by Gender

In Table 4, remarkable sex differences in body composition could be observed. FFMI (17.89 ± 1.75 vs. 14.90 ± 1.43) was significantly lower in females compared with males (*p* < 0.001). Contrarily, women had higher values of VFA (56.33 ± 29.40 vs. 70.96 ± 21.19), WHtR (0.46 ± 0.06 vs. 0.48 ± 0.05), FMI (4.73 ± 2.62 vs. 7.18 ± 2.51), and BFP (20.00 ± 6.83 vs. 31.72 ± 6.06) than males (all *p* < 0.005). In addition, Appendix A showed body composition differences in demographic characteristics and Appendix A showed body composition differences in living habits characteristics.

### 3.4. Multiple Linear Regression Analyses between Sleep Characteristics and Body Composition Indicators

In multivariate linear regression analysis (Table 5, Figure 1), after adjusting for gender, father’s education, mother’s education, breakfast consumption, fruits consumption, pure juice consumption (>250 mL), soft drinks consumption (>250 mL), alcohol consumption, smoking, duration of physical exercise each time, individuals with weekday nap duration/d (>30 min) had higher WHtR (*B* = 0.013, FDR-corrected *p* = 0.080) (Figure 1c). Compared with the average sleep duration (>7h/d), participants with the average sleep duration (≤7 h/d) showed a significant increase of 1.117 kg/m^2^ in BMI (FDR-corrected *p* = 0.176) (Figure 1a) and 0.016 in WHtR (FDR-corrected *p* = 0.080) (Figure 1c). There was also a trend for adolescents with great social jetlag (>1 h) to have more VFA (*B* = 7.475), WHtR (*B* = 0.015), WHR (*B* = 0.012), FMI (*B* = 0.663) and BFP (*B* = 1.703) (all FDR-corrected *p* < 0.1). Spending more screen time before sleep (>0.5 h) was associated with an increased risk for having higher VFA (*B =* 7.934, FDR-corrected *p* = 0.064) (Figure 1b), WHtR (*B* = 0.017, FDR-corrected *p* = 0.080) (Figure 1c), WHR (*B* = 0.016, FDR-corrected *p* = 0.090) (Figure 1c), FMI (*B* = 0.902, FDR-corrected *p* = 0.069) (Figure 1e) and BFP (*B* = 2.892, FDR-corrected *p* = 0.018) (Figure 1g). However, no other obvious associations were noted across sleep characteristics and body-composition-related indexes.

## 4. Discussion

In the current study, a significant association was observed between poor sleep behaviors and general and abdominal obesity. After adjusting for covariates, weekday nap duration (>30 min/d) was positively related to higher WHtR values. Adolescents with average sleep duration (≤7 h/d) showed obvious increases in WHtR. High SJL could positively affect general and abdominal obesity (observed by VFA, WHtR, WHR, FMI and BFP). More screen time before sleep (>0.5 h) also exhibited higher risk of obesity by VFA, WHtR, WHR, FMI and BFP. To the best of our knowledge, this is the first cross-sectional study to investigate the association between various representative sleep characteristics with comprehensive body composition indicators measured by BIA in Chinese adolescents.

Our findings regarding average sleep duration were consistent with some previous literature. For instance, a systematic research suggested that short sleep duration was associated with the risk of developing overweight or obesity among Chinese children and adolescents [42]. A study in China with 9059 participants (63.08% were females) revealed that short sleep duration caused the elevated rates of general obesity (defined using BMI) and visceral obesity with sex differences [43]. The results acquired in Mexican children/adolescents exposed that the combined effects of inactivity, excessive screen time and inadequate sleep led to overweight/obesity and higher values of FM [44]. Several mechanisms have been proposed. Hormonal changes are thought to be involved in the relationship between sleep duration and obesity. Leptin is able to suppress appetite, while ghrelin has opposing functions of stimulating hunger [44]. A clinical research including 12 healthy young men demonstrated that when sleep time was restricted to 4 h sleeping in bed at night over two days, subjects’ leptin level decreased by 18% and ghrelin level increased by 28% compared with hormonal levels of those spending 10 h in bed [45]. However, some studies also demonstrated that between sleep restriction and normal sleep, the former had no significant effect on ghrelin and leptin levels [46,47]. While the underlying explanations need to be further explored, it is clear that short sleep duration brings daytime sleepiness and fatigue, which may result in a low physical activity [47]. Most likely, no separate factor explains the impact of short sleep on obesity completely, but multiple mechanisms work together. Conversely, obstructive sleep apnea frequently occurs among obese patients [48] and patients can have an improvement with weight loss [49]. Therefore, bidirectional relationships may link insufficient sleep duration to obesity [50,51].

Our results supported that high SJL had a positive impact on both general and abdominal obesity, which was in line with several previous literature. Serving as a measure of circadian misalignment, SJL has played a role as a risk factor in obesity-related chronic diseases [51] and was significantly associated with an increased WC in a Japanese working population [52]. The explanatory mechanisms remain unclear. Some hypothesized that altered sleep–wake cycles could influence obesogenic environments of adipocytes and the central circadian clock by modifying gene expression [39,53]. Emerging evidence indicates that circadian misalignment leads to increased weight and the development of obesity [54]. Apart from this, high SJL has associations with unhealthy dietary patterns, which are likely to cause overweight or obesity. According to a study of 3060 adolescents in the United States, high SJL led to more choices to sweetened beverages and fast food, which caused obesity (defined using BMI) and other negative health outcomes [55]. Among 534 young adults (18–25 years), individuals with greater SJL were related to a lesser intake of fruits and vegetables, even skipping breakfast [56]. Consuming a lower quantity of the recommended amount of vegetables and fruits increased the rise of overweight and obesity [57] as evidence suggests that bioactive compounds present in vegetables and fruits, containing carotenoids, polyphenols, and phytosterols, possibly have beneficial associations with the prevention of obesity and other diseases [58].

Our results revealed that weekday nap duration (≤30 min/d) was inversely related to WHR, which was consistent with previous studies including in Southwest China [52], Iran [59] and among Latino populations [60]. Several hypotheses may be proposed. First, daytime naps can stimulate the sympathetic nervous system, which is positive with obesity [61,62], but the specific mechanism of action seems to be unclear. Second, a long nap duration may cause E-Type and more sleep latency, even insomnia at night, which are obviously relevant with obesity.

Media use such as mobile phone, computer or television, etc., and increasing screen time before sleep or even after lights out have become a common practice, especially among college students. In the current study, nearly 90% of the participants had screen time before sleep for more than 0.5 h, an astonishing proportion. Our results indicated that screen time before sleep (>0.5 h) led to more potential to general and abdominal obesity (higher VFA, BFP, WHR, WHtR, etc.), which was consistent with previous findings. A study on Indian adolescents exposed excess screen time could affect adiposity indicators [63]. Although the exact underlying mechanisms remain unclear, we hypothesize that various amounts of blue light emitted by electronic screens could suppress melatonin production [64] and might be deleterious to sleep cycles. Studies regarding adolescents have demonstrated that screen time during bedtime has impacts on poor sleep outcomes (shorter sleep duration, longer sleep latency, lower sleep efficiency, etc.) [65,66], which can influence biological and social rhythms (e.g., evening chronotype, SJL) or circadian rhythms [64,67]. Interfering with excess screen time in bed and avoiding sleep disorders is a practical step to prevent obesity among adolescents effectively.

Several strengths of our study deserve mentioning. The use of InBody 720 captures non-invasive and highly precise assessments of fat distribution and body composition, during a short measurement time. In addition, the selected indicators were representative enough to describe both general and abdominal obesity. Third, not only did we concentrate on different dimensions of sleep characteristics, but also we attached great importance to the weekday-to-weekend sleep differences in this special population such as sleep midpoint, SJL, etc. With the increasing popularity of electronic devices among young individuals, we focused on screen time before sleep to analyze the association with general and abdominal adiposity. Additionally, all descriptive analyses were stratified by gender and accordingly we controlled confounders including sex in linear regression analyses. We controlled for various kinds of potential confounding factors, including demographic variables, dietary intake, alcohol, smoking and physical exercise.

Some limitations of this study should be highlighted. First, owing to the cross-sectional design of our analysis, causal relationships could not be allowed. The participants were selected from one university, which made selection bias unavoidable. Thus, futural investigations should contain well-designed longitudinal studies and cover populations from different regions. Besides, instead of using objective measurements of sleep characteristics such as accelerometers, we applied subjective questionnaires. However, it seemed that our approach was good enough to achieve our purpose among participants. Furthermore, psychological factors such as depression and mood disturbances are probably related to both obesity and sleep. Thus, ongoing studies should include more influential factors to verify the results. Moreover, standing position was used in the BIA (InBody 720), which might bring a few inaccuracies of the measured parameters. To prefer the measurement of body composition in the future study, before contacting with the electrodes, participants should be guided to clean their hands and feet with antibacterial tissues provided by the manufacturer. Finally, we did not consider genetic markers of obesity, possibility of familial obesity inheritance of participants, and factors or conditions related to sleep disturbance, including sleep apnea, insomnia, etc.

## 5. Conclusions

In this study, we showed that poor sleep characteristics were involved in the development of general and abdominal obesity among Chinese adolescents. Different sleep behaviors could have various associations with body composition parameters. To this end, we consider that encouraging adolescents to follow recommended sleep protocols and promoting good sleep behaviors is a promising and practical choice to prevent obesity or other chronic diseases. Further explorations can focus on the relationship to explore more clear mechanisms, develop appropriate strategies against obesity and relevant metabolic disorders.

## Figures and Tables

**Figure 1 nutrients-15-04130-f001:**
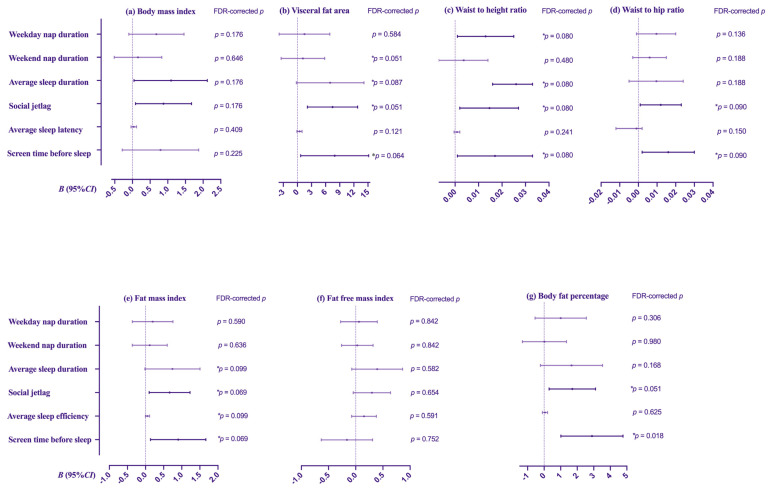
Multiple linear regression analyses between sleep characteristics and body composition. Models were adjusted for gender, father’s education, mother’s education, breakfast consumption, fruits consumption, pure juice consumption, soft drinks consumption, alcohol consumption, smoking, duration of physical exercise each time. * Indicates that after the Benjamini–Hochberg false discovery rate (FDR) correction, the *p* value < 0.1.

**Table 1 nutrients-15-04130-t001:** Demographic characteristics of the study population stratified by sex.

Variables	All	Males	Females	*χ* ^2^	*p*
Race				0.113	0.737
Han	423 (95.3)	146 (94.8)	277 (95.5)		
Minority	21 (4.7)	8 (5.2)	13 (4.5)		
Location				0.008	0.931
Rural	290 (65.3)	101 (65.6)	189 (65.2)		
Urban	154 (34.7)	53 (34.4)	101 (34.8)		
The only child				2.331	0.127
No	355 (80.0)	117 (76.0)	238 (82.1)		
Yes	89 (20.0)	37 (24.0)	52 (17.9)		
Father’s education				5.246	0.263
Below elementary school	29 (6.5)	14 (9.1)	15 (5.2)		
Elementary school	45 (10.1)	12 (7.8)	33 (11.4)		
Junior school	166 (37.4)	62 (40.3)	104 (35.9)		
High school or technical secondary school	107 (24.1)	32 (20.8)	75 (25.9)		
Junior college or above	97 (21.8)	34 (22.1)	63 (21.7)		
Mother’s education				2.812	0.590
Below elementary school	58 (13.1)	24 (15.6)	34 (11.7)		
Elementary school	65 (14.6)	20 (13.0)	45 (15.5)		
Junior school	131 (29.5)	42 (27.3)	89 (30.7)		
High school or technical secondary school	104 (23.4)	40 (26.0)	64 (22.1)		
Junior college or above	86 (19.4)	28 (18.2)	58 (20.0)		
Self-rated family income				2.804	0.246
Low	147 (33.1)	53 (34.4)	94 (32.4)		
Middle	270 (60.8)	88 (57.1)	182 (62.8)		
High	27 (6.1)	13 (8.4)	14 (4.8)		

Note: data are expressed as number (%).

**Table 2 nutrients-15-04130-t002:** Living habits characteristics of the study population stratified by sex.

Variables	All	Males	Females	*χ* ^2^	*p*
Takeaway food consumption/week				14.944	**0.001**
0	141 (31.8)	42 (27.3)	99 (34.1)		
1–2 servings	188 (42.3)	53 (34.4)	135 (46.6)		
>2 servings	115 (25.9)	59 (38.3)	56 (19.3)		
Breakfast consumption/week				5.369	**0.020**
<7 days	276 (62.2)	107 (69.5)	169 (58.3)		
7 days	168 (37.8)	47 (30.5)	121 (41.7)		
Vegetables consumption/d				0.737	0.391
<3 servings	342 (77.0)	115 (74.7)	227 (78.3)		
≥3 servings	102 (23.0)	39 (25.3)	63 (21.7)		
Fruits consumption/d				16.067	**<0.001**
<1 servings	210 (47.3)	88 (57.1)	122 (42.1)		
1 serving	126 (28.4)	42 (27.3)	84 (29.0)		
>1 servings	108 (24.3)	24 (15.6)	84 (29.0)		
Dried fruits consumption/d				1.400	0.497
0	238 (53.6)	82 (53.2)	156 (53.8)		
<1 servings	140 (31.5)	45 (29.2)	95 (32.8)		
≥1 servings	66 (14.9)	27 (17.5)	39 (13.4)		
Pure juice consumption (>250 mL)/d				11.307	**0.004**
0	289 (65.1)	82 (53.2)	207 (71.4)		
<1 servings	115 (25.9)	48 (31.2)	67 (23.1)		
≥1 servings	40 (9.0)	24 (15.6)	16 (5.5)		
Soft drinks consumption (>250 mL)/week				15.286	**<0.001**
0	177 (39.9)	44 (28.6)	133 (45.9)		
1 serving	138 (31.1)	46 (29.9)	92 (31.7)		
>1 servings	129 (29.1)	64 (41.6)	65 (22.4)		
Sugar-sweetened beverage consumption (>250 mL)/week				3.129	0.209
0	111 (25.0)	38 (24.7)	73 (25.2)		
1 serving	140 (31.5)	44 (28.6)	96 (33.1)		
>1 servings	193 (43.5)	72 (46.8)	121 (41.7)		
Alcohol consumption				26.115	**<0.001**
No	391 (88.1)	119 (77.3)	272 (93.8)		
Yes	53 (11.9)	35 (22.7)	18 (6.2)		
Smoking				7.798	**0.005**
No	436 (98.2)	147 (95.5)	289 (99.7)		
Yes	8 (1.8)	7 (4.5)	1 (0.3)		
Duration of physical exercise each time				13.190	**<0.001**
<60 min	411 (92.6)	133 (86.4)	278 (95.9)		
≥60 min	33 (7.4)	21 (13.6)	12 (4.1)		
Number of physical exercise/d				2.890	0.089
<1	371 (83.6)	135 (87.7)	236 (81.4)		
≥1	73 (16.4)	19 (12.3)	54 (18.6)		
Weekday screen time/d				4.031	0.133
<2 h	121 (27.3)	32 (20.8)	89 (30.7)		
<4 h	192 (43.2)	75 (48.7)	117 (40.3)		
≥4 h	131 (29.5)	47 (30.5)	84 (29.0)		
Weekend screen time/d				15.653	**0.016**
<2 h	73 (16.4)	27 (17.5)	46 (15.9)		
<4 h	159 (35.8)	50 (32.5)	109 (37.6)		
≥4 h	212 (47.7)	77 (50.0)	135 (46.6)		

Note: data are expressed as number (%). Bold: *p* < 0.05.

**Table 3 nutrients-15-04130-t003:** Sleep variables of the study population stratified by sex.

Variables	All	Males	Females	*χ^2^/t*	*p*
Weekday nap duration/d				13.852	**<0.001**
≤30 min	330 (74.3)	97 (63.0)	233 (80.3)		
>30 min	114 (25.7)	57 (37.0)	57 (19.7)		
Weekend nap duration/d				0.338	0.561
≤30 min	234 (52.7)	79 (51.3)	155 (53.4)		
>30 min	210 (47.3)	75 (48.7)	135 (46.6)		
Average sleep duration/d				0.102	0.749
≤7 h	49 (11.0)	18 (11.7)	31 (10.7)		
>7 h	395 (89.0)	136 (88.3)	259 (89.3)		
Social jetlag				5.846	**0.016**
≤1 h	333 (75.0)	105 (68.2)	228 (78.6)		
>1 h	111 (25.0)	49 (31.8)	62 (21.4)		
Average sleep efficiency (%)/d	95.2 ± 4.3	95.2 ± 4.7	95.3 ± 4.0	0.335	0.738
Screen time before sleep (>0.5 h)				0.043	0.836
No	50 (11.3)	18 (11.7)	32 (11.0)		
Yes	394 (88.7)	136 (88.3)	258 (89.0)		

Note: data are expressed as number (%) and mean± SD. SD: standard deviation. Bold: *p* < 0.05.

**Table 4 nutrients-15-04130-t004:** Body composition indicators of the study population stratified by sex.

Variables	All	Males	Females	*t*	*p*
Visceral fat area (cm^2^)	65.89 ± 25.30	56.33 ± 29.40	70.96 ± 21.19	−5.468	**<0.001**
Body mass index (kg/m^2^)	22.29 ± 3.58	22.65 ± 3.78	22.10 ± 3.46	1.536	0.125
Waist to height ratio	0.47 ± 0.06	0.46 ± 0.06	0.48 ± 0.05	−3.221	**0.001**
Waist to hip ratio	0.83 ± 0.05	0.83 ± 0.06	0.83 ± 0.04	0.337	0.737
Fat mass index	6.33 ± 2.80	4.73 ± 2.62	7.18 ± 2.51	−9.637	**<0.001**
Fat free mass index	15.94 ± 2.10	17.89 ± 1.75	14.90 ± 1.43	18.284	**<0.001**
Body fat percentage (%)	27.65 ± 8.44	20.00 ± 6.83	31.72 ± 6.06	−18.554	**<0.001**

Note: data are expressed as mean ± SD. SD: standard deviation. Bold: *p* < 0.05.

**Table 5 nutrients-15-04130-t005:** Association of body composition parameters with different sleep characteristics of study population.

**Variables**	**Visceral Fat Area (cm^2^)**				**Body Mass Index (kg/m^2^)**				**Waist to Height Ratio**				**Waist to Hip Ratio**			
	**Model 1 B (95%CI)**		**Model 2 B (95%CI)**		**Model 1 B (95%CI)**		**Model 2 B (95%CI)**		**Model 1 B (95%CI)**		**Model 2 B (95%CI)**		**Model 1 B (95%CI)**		**Model 2 B (95%CI)**	
Weekday nap duration/d	1.258	(−4.148, 6.664)	1.500	(−3.885, 6.884)	0.879	**(0.119, 1.640)**	0.678	(−0.102, 1.458)	0.011	(<0.001, 0.023)	0.013	**(0.001, 0.025)**	0.011	**(0.001, 0.022)**	0.010	(−0.001, 0.020)
Weekend nap duration/d	0.964	(−3.765, 5.692)	1.181	(−3.451, 5.812)	0.250	(−0.419, 0.918)	0.158	(−0.515, 0.830)	0.005	(−0.006, 0.015)	0.004	(−0.007, 0.014)	0.007	(−0.002, 0.016)	0.006	(−0.003, 0.015)
Average sleep duration	7.338	(−0.170, 14.846)	6.963	(−0.227, 14.152)	1.172	**(0.112, 2.232)**	1.117	**(0.045, 2.188)**	0.017	**(<0.001, 0.033)**	0.016	**(0.029, 0.033)**	0.011	(−0.004, 0.025)	0.010	(−0.005, 0.024)
Social jetlag	4.860	(−0.577, 10.296)	7.475	**(2.137, 12.813)**	0.838	**(0.071, 1.605)**	0.878	**(0.086, 1.671)**	0.011	(−0.001, 0.023)	0.015	**(0.002, 0.027)**	0.012	**(0.002, 0.022)**	0.012	**(0.001, 0.023)**
Average sleep efficiency/d	0.386	(−0.167, 0.940)	0.449	(−0.088, 0.987)	0.035	(−0.044, 0.113)	0.038	(−0.040, 0.116)	0.001	(−0.001, 0.002)	0.001	(<0.001, 0.002)	0.001	(<0.001, 0.002)	0.001	(<0.001, 0.002)
Screen time before sleep (>0.5 h)	6.032	(−1.420, 13.483)	7.934	**(0.700, 15.167)**	0.568	(−0.487, 1.624)	0.794	(−0.288, 1.876)	0.015	(−0.001, 0.031)	0.017	**(0.001, 0.033)**	0.015	**(0.001, 0.029)**	0.016	**(0.002, 0.030)**
**Variables**	**Fat mass index**						**Free fat mass index**					**Body fat percentage (%)**				
	**model 1 B (95%CI)**				**model 2 B (95%CI)**		**model 1 B (95%CI)**			**model 2 B (95%CI)**		**model 1 B (95%CI)**			**model 2 B (95%CI)**	
Weekday nap duration/d	0.267		(−0.327, 0.861)		0.196	(−0.364, 0.757)	0.673	**(0.127, 1.019)**		0.060	(−0.277, 0.397)	0.945	(−0.858, 2.747)		1.003	(−0.545, 2.551)
Weekend nap duration/d	0.069		(−0.451, 0.588)		0.116	(−0.366, 0.599)	0.099	(−0.293, 0.492)		0.029	(−0.260, 0.319)	0.201	(−1.378, 1.779)		0.017	(−1.317, 1.351)
Average sleep duration	0.747		(−0.084, 1.577)		0.743	(−0.018, 1.503)	0.452	(−0.173, 1.078)		0.395	(−0.072, 0.862)	1.632	(−0.878, 4.142)		1.655	(−0.223, 3.534)
Social jetlag	0.289		(−0.313, 0.892)		0.663	**(0.099, 1.228)**	0.616	**(0.166, 1.066)**		0.297	(−0.046, 0.640)	0.094	(−1.725, 1.914)		**1.703**	**(0.301, 3.105)**
Average sleep efficiency/d	0.045		(−0.016, 0.105)		0.052	(−0.003, 0.108)	0.011	(−0.035, 0.057)		0.152	(−0.076, 0.380)	0.025	(−0.160, 0.210)		0.051	(−0.104, 0.206)
Screen time before sleep (>0.5 h)	0.734		(−0.090, 1.557)		0.902	**(0.138, 1.666)**	−0.193	(−0.814, 0.428)		−0.161	(−0.633, 0.310)	2.464	(−0.018, 4.946)		2.892	**(1.014, 4.771)**

Model 1: A simple linear regression analysis of the sleep variables and parameters on body composition was conducted after no adjustment. Model 2: A multiple linear regression analysis of the sleep variables and parameters on body compositions was conducted after adjustment for gender, father’s education, mother’s education. breakfast consumption, fruits consumption, pure juice consumption, soft drinks consumption, alcohol consumption, smoking, and duration of physical exercise each time. Bold: *p* < 0.05.

## Data Availability

The datasets used and/or analyzed during the current study are available from the corresponding author on reasonable request.

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
