# Peer review of "Sleep-Body Composition Relationship: Roles of Sleep Behaviors in General and Abdominal Obesity in Chinese Adolescents Aged 17–22 Years"

_nutrients, 2023, doi:10.3390/nu15194130_

Round 1
Reviewer 1 Report
The study focused on the interesting topic of sleep quality and body composition. Thank you for the possibility to review this article. This is a very important issue, however, I would like to express my concerns about the parts of the manuscript that need corrections, elaborating, explanation, or general improvement.
The manuscript requires extensive English revision including the title of the paper – I suggest using the term relationship instead of relation.
Abstract line 18 – What is hipline or abdominal circumference? Please use proper and relevant professional terms instead of colloquial ones.
Please provide also the p-value for the presented results.
Line 32 – Please describe these criteria.
Line 59 – Please describe details of these types.
Line 73 and 77 – There are also other important factors impacting sleep quality in young adults, especially after pandemics, e.g. knowledge or supplementation – please mention also these factors.
Line 100 – Were the students coming from one specific department that could impact their sleep habits?
Line 104 – Please describe inclusion and exclusion criteria in detail. Were any medical conditions that could impact obesity acknowledged as exclusion criteria?
Line 122 – Were the BIA measurements taken in the morning or afternoon (the measurement should be taken with the producer’s instructions regarding the interval from the last meal).
The BIA measurement in standing position brings some inaccuracies of the measured parameters – the Authors should indicate these limitations.
Please describe how the waist circumference was measured.
Table 2 – what does “takeaway food consumption/week” mean, when the answer is yes/no? Did the authors mean no = 0 yes = any, or were there any cut-off values?
Is the same for fruits?
In general, what period of time were the respondents asked about regarding their habits? Last week/year?
Line 242-262 – please present this data in a table to make it easier for the readers.
Line 315-316 – Please elaborate on that topic. Why is this observation important, are there any compounds present in vegetables related to obesity prevention?
The manuscript requires extensive English revision including the title of the paper – I suggest using the term relationship instead of relation.
Author Response
Dear Editors and Reviewers:
We would like to thank you for giving us a chance to revise our manuscript entitled “Sleep-body Composition Relation: Roles of Sleep Behaviors in General and Abdominal Obesity in Chinese College Students (ID: nutrients-2547197)” by Song et al. We also thank the reviewers for their valuable comments and suggestions to improve the quality of our manuscript. We have revised the manuscript accordingly. A point-by-point response to the reviewer’s comments are listed below.
Reviewer #1:
The study focused on the interesting topic of sleep quality and body composition. Thank you for the possibility to review this article. This is a very important issue, however, I would like to express my concerns about the parts of the manuscript that need corrections, elaborating, explanation, or general improvement.
Q1. The manuscript requires extensive English revision including the title of the paper – I suggest using the term relationship instead of relation.
Response: We are grateful to the reviewer for his/her constructive suggestions.
In the revised manuscript, we modified the title: “Sleep-body Composition Relationship: Roles of Sleep Behaviors in General and Abdominal Obesity in Chinese Adolescents.” (Page 1, lines 2-4, marked in blue)
We tried our best to polish the language and made extensive revision of the manuscript. Here we didn’t list these changes, but they were marked in blue in the revised manuscript
Q2. Abstract line 18 – What is hipline or abdominal circumference? Please use proper and relevant professional terms instead of colloquial ones.
Please provide also the p-value for the presented results.
Response: We are thankful to the review for his/her constructive suggestions and comments.
Hipline is defined the horizontal circumference of the most prominent part of the buttocks. Abdominal circumference refers to the horizontal circumference of the abdomen through the iliac crest point and waist circumference refers to the circumference of the midpoint line between the lowest point of the rib and the upper edge of the iliac crest(Nie et al., 2020). In order to describe body composition indicators more accurately and avoid confusing readers, we have decided to replace abdominal circumference with the professional term waist circumference, and use the term hip circumference instead of hipline. We are sorry for not using professional terms properly. Additionally, we agree with and appreciate the reviewer’s proposal, providing the p-value accordingly in the revised manuscript.
In the Abstract, the sentences were modified:
“Regression models tested relationships between sleep and body composition after adjustment for covariates. Students with weekday nap duration (>30min/d) exerted higher waist-height ratio (WHtR) (B=0.013, FDR-corrected P=0.080). Average sleep duration (≤7h/d) was linked to more WHtR (B=0.016, FDR-corrected P=0.080). People with high social jetlag showed gained VFA (B=7.475), WHtR (B=0.015), WHR (B=0.012), FMI (B=0.663) and BFP (B=1.703) (all FDR-corrected P<0.1). Individuals with screen time before sleep (>0.5 h) exhibited higher visceral fat area (B=7.934, FDR-corrected P=0.064), WHtR (B=0.017, FDR-corrected P=0.080), waist to hip ratio (B=0.016, FDR-corrected P=0.090), fat mass index (B=0.902, FDR-corrected P=0.069) and body fat percentage (B=2.892, FDR-corrected P=0.018).” (Page 1, lines 14-23, marked in blue)
In the Introduction, the sentence was supplemented:
“For instance, visceral adipose tissue is known to be more closely linked to cardiovascular risk than subcutaneous adipose tissue, thus it is more beneficial to assess waist circumference (WC) or waist to hip ratio (WHR) instead of BMI [13].” (Page 2, lines 51-54, marked in blue)
In the Materials and Methods, we modified the sentences:
“Weight, hip circumference (HC), FM, FFM, muscle mass, visceral fat area (VFA), BMI, WHtR, WHR and BFP were assessed by BIA using Inbody 720.” (Page 4, lines 164-166, marked in blue)
The following reference was supplemented in the Reference List (marked in blue):
- Stich FM, Huwiler S, D'Hulst G, Lustenberger C: The Potential Role of Sleep in Promoting a Healthy Body Composition: Underlying Mechanisms Determining Muscle, Fat, and Bone Mass and Their Association with Sleep. Neuroendocrinology 2022, 112(7):673-701.
Q3. Line 32 – Please describe these criteria.
Response: We thank the reviewer for his/her insightful suggestions and comments.
We apologize for not being able to provide a detailed description of the criteria. According to WHO criteria, overweight is defined as a BMI of 25.0–29.9 kg/m² and obesity as a BMI of 30.0 kg/m² or higher in adults with no sex differences (2000, Pan et al., 2021). However, evidence has shown that Chinese people seem to have higher body fat percentages (Pan et al., 2021). Supported by accumulated evidence from the China Kadoorie Biobank, there are different criteria recommended by The Working Group on Obesity in China, respectively setting BMI cutoffs of 24.0 kg/m² to define overweight and 28.0 kg/m² to define obesity with no sex differences. BMI standards for individuals (<18 years) are divided by age group, which are not uniform and more complex with gender differences (Gao et al., 2019, Li et al., 2019).
In the Introduction, the sentence was supplemented:
“Based on Chinese criteria, overweight is defined as a BMI of 24.0 kg/m² and obesity as a BMI of 28.0 kg/m² or higher in adults (≥18 years) [11.12].” (Page 2, lines 47-49 marked in blue)
The following reference was supplemented in the Reference List (marked in blue):
- Gao M, Wei YX, Lyu J, Yu CQ, Guo Y, Bian Z, Pei P, Du HD, Chen JS, Chen ZM et al: [The cut-off points of body mass index and waist circumference for predicting metabolic risk factors in Chinese adults]. Zhonghua Liu Xing Bing Xue Za Zhi 2019, 40(12):1533-1540.
- Pan XF, Wang L, Pan A: Epidemiology and determinants of obesity in China. Lancet Diabetes Endocrinol 2021, 9(6):373-392.
Q4. Line 59 – Please describe details of these types.
Response: We thank the reviewer for his/her valuable comments and suggestions.
We apologize for failing to provide the details of the types. Chronotype refers to individual preferences in the timing of daily activities, usually evaluated with the Morningness-Eveningness Questionnaire (MEQ) (Mongrain et al., 2004), which is classified as: Morning-type (M-type), Evening-type (E-type) and Neither-type (N-type) (Montaruli et al., 2021). People who belong to M-type go to bed and wake up early, doing their peak mental and physical performance in the early part of the day. By contrast, E-type individuals have a later bedtime/wake-up time, reaching their best achievements during the second half of the day (Adan et al., 2012). Participants with no noticeable circadian preference are regarded as N-type as they show intermediate characteristics (Montaruli et al., 2021).
Unfortunately, we have removed several sleep variables in the revised manuscript. Ultimately, we chose to keep social jetlag as the sleep characteristic related to endogenous circadian rhythm.
Q5. Line 73 and 77 – There are also other important factors impacting sleep quality in young adults, especially after pandemics, e.g. knowledge or supplementation – please mention also these factors.
Response: We agree with the reviewer’s comment and suggestion.
The coronavirus disease 2019 (COVID-19) pandemic, which swept large parts of the world, has exacerbated the prevalence of sleep problems (e.g. shorter sleep duration, poor sleep quality, misaligned sleep midpoint, etc.)in all populations, especially the groups of children and adolescents (Catarozoli, 2023, Jahrami et al., 2022).
In the Introduction, the sentence was supplemented:
“Of note, the COVID-19 pandemic has exacerbated the prevalence of sleep problems, which may contribute to shorter sleep duration, poor sleep quality, low sleep efficiency, etc. in all populations, especially the groups of children and adolescents [19,20].” (Page 2, lines64-67 marked in blue)
The following reference was supplemented in the Reference List (marked in blue):
- Catarozoli C: Sleep During the Pandemic. Sleep Med Clin 2023, 18(2):219-224.
- Jahrami HA, Alhaj OA, Humood AM, Alenezi AF, Fekih-Romdhane F, AlRasheed MM, Saif ZQ, Bragazzi NL, Pandi-Perumal SR, BaHammam AS et al: Sleep disturbances during the COVID-19 pandemic: A systematic review, meta-analysis, and meta-regression. Sleep Med Rev 2022, 62:101591.
Q6. Line 100 – Were the students coming from one specific department that could impact their sleep habits?
Response: We thank the reviewer for his/her valuable comments and suggestions.
The students selected from one specific department could make selection bias unavoidable, which caused insufficient representativeness and universality, making it difficult to accurately reflect the relationship between sleep characteristics and body composition in Chinese adolescents.
In our study, a total of 469 students from freshman to senior, aged 17-22 years, from the School of Pharmacy, School of Public Health and School of Stomatology of Zhengzhou University (Zhengzhou, Henan, China) were recruited to participate in this cross-sectional research. In futural study, we should cover populations from more regions and collect more representative data.
In the Materials and Methods, the sentence was supplemented and modified:
“A total of 469 healthy students from freshman to senior, aged 17-22 years, from the School of Pharmacy, School of Public Health and School of Stomatology of Zhengzhou University (Zhengzhou, Henan, China) were recruited using a convenience sample in this cross-sectional observational study.” (Page 3, lines113-116 marked in blue)
Q7. Line 104 – Please describe inclusion and exclusion criteria in detail. Were any medical conditions that could impact obesity acknowledged as exclusion criteria?
Response: We thank the reviewer for his/her insightful and constructive suggestions.
We are sorry for the rough inclusion and exclusion criteria about the sample. Inclusion criteria consisted of healthy university students aged 17-22 years, having all completed and valid data. Exclusion criteria were contraindications to BIA, such as diseases affecting the electrical resistance of the skin, pregnancy, having an implanted pacemaker, and the presence of metal prostheses. Individuals with endocrine or renal diseases such as type 2 diabetes, cancer or sever inflammatory conditions were not included in the sample. Participants who did not have complete general characteristics information were excluded (n=15). We excluded 3 adolescents whose midpoints of sleep during weekdays were between 22:00 and 24:00 because their data was invalid, and could have resulted in a lack of statistical significance. 7 individuals did not have complete questionnaires data related to sleep. All in all, 25 participants were excluded from the analysis, and 444 students met the criteria.
For the purpose of investigating the association between sleep behaviors and body composition in Chinese adolescents, it was necessary to consider some medical conditions that possibly impacted obesity determination such as sever inflammatory conditions, endocrine or renal diseases.
In the Materials and Methods, the sentences were supplemented and modified:
“A total of 469 healthy students from freshman to senior, aged 17-22 years, from the School of Pharmacy, School of Public Health and School of Stomatology of Zhengzhou University (Zhengzhou, Henan, China) were recruited using a convenience sample in this cross-sectional observational study. Exclusion criteria were contraindications to BIA, such as diseases affecting the electrical resistance of the skin, pregnancy, having an implanted pacemaker, and the presence of metal prostheses. Individuals with endocrine or renal diseases such as type 2 diabetes, cancer or sever inflammatory conditions were not included in the sample. In the study, due to the application of BIA, all students had exact body composition data. Participants who did not have complete general characteristics information were excluded (n=15). We excluded 3 adolescents whose midpoints of sleep during weekdays were between 22:00 and 24:00 because their datza were invalid, and could have resulted in a lack of statistical significance. 7 individuals did not have complete related sleep questionnaires data. Overall, 25 participants were excluded from the analysis, and 444 students met the criteria.” (Page 3, lines113-126 marked in blue)
Q8. Line 122 – Were the BIA measurements taken in the morning or afternoon (the measurement should be taken with the producer’s instructions regarding the interval from the last meal).
Response: We thank the reviewer for his/her valuable comments and suggestions.
In our study, the BIA measurements, which were used to assess body composition, started at about 7:00 p.m. and lasted for an hour and a half. Students were instructed not to drink or eat, and to void their bladder before the evaluations. Therefore, this measurement time ensured an appropriate and sufficient interval from last dinner.
In the Materials and Methods, the sentences were supplemented and modified:
“Additionally, anthropometric and bioelectrical measurements were carried out at about 7:00 p.m. and lasted for an hour and a half every day at an appropriate space from November 8, 2021 to November 21, 2021.” (Page 3, lines131-134 marked in blue)
Q9. The BIA measurement in standing position brings some inaccuracies of the measured parameters – the Authors should indicate these limitations.
Response: We are appreciated for the reviewer’s comments and suggestions.
With the invention of various bioelectric impedance analysis tools, the subject's position during the test, such as lying, sitting and standing, possibly is of vital importance for the comparability of the results. There were significant differences in reactance, phase angle, body cell mass, intracellular and extracellular water measurements between lying and standing, sitting and standing (Wiech et al., 2022). The InBody S10 is widely used for hemodialysis patients in the lying position as the distribution of body fluid is stable in this position (Choi et al., 2022). However, the estimated weight, fat mass, etc., were consistent between the lying position using the InBody S10 and the standing position using the InBody 770 (Choi et al., 2022).
The Inbody analyzer was applied in our study (InBody 720, Biospace Co. Ltd., Seoul, Republic of Korea) using a standing position to assessed weight, hip circumference, waist circumference, fat mass, lean mass, visceral fat area, body mass index and body fat percentage. There might be a few inaccuracies of the measured parameters. Therefore, in the future study, before contacting with the electrodes, participants should be guided to clean their hands and feet with antibacterial tissues provided by the manufacturer (McLester et al., 2020).
In the Discussion, following sentences were supplemented:
“Moreover, standing position was used in the BIA (InBody 720, Biospace Co. Ltd., Seoul, Republic of Korea), which might bring a few inaccuracies of the measured parameters. To prefer the measurement of body composition in the future study, before contacting with the electrodes, participants should be guided to clean their hands and feet with antibacterial tissues provided by the manufacturer.” (Page 13, lines406-411 marked in blue)
Q10. Please describe how the waist circumference was measured.
Response: We are appreciated for the reviewer’s comments and suggestions.
Waist circumference was measured using a non-elastic tape at the midpoint line between the lowest point of the rib and the upper edge of the iliac crest, to the nearest 0.1 cm. In our study, although waist circumference could be measured by bioelectrical impedance analysis (BIA) using the InBody 720, we chose a more precise way with anon-elastic tape to measure it.
In the Materials and Methods, the sentence was supplemented:
“Waist circumference was measured using a non-elastic tape at the midpoint line between the lowest point of the rib and the upper edge of the iliac crest, to the nearest 0.1 cm.” (Page 3, lines135-137 marked in blue)
Q11. Table 2 – what does “takeaway food consumption/week” mean, when the answer is yes/no? Did the authors mean no = 0 yes = any, or were there any cut-off values?
Is the same for fruits?
In general, what period of time were the respondents asked about regarding their habits? Last week/year?
Response: We thank the reviewer for his/her insightful suggestions and comments.
In our study, “takeaway food consumption/week” was evaluated by the question “In the last week, how many servings did you have takeaway food on average?” with response options of: “none”, “1-2”, “2-4” and “>4”. Subsequently, we merged the last three options, which means “yes”, and binarized the answer as yes/no. Any number of consumption greater than zero was considered “yes”.
The evaluation method for “fruits consumption/d” was similar to the above. Subjects were asked “In the last week, how many servings did you have fruits on average per day” with response options of: “none”, “0-1” and “>1”. We also adopted the method of binary answers as yes/no.
To make the description of living habits more detailed and avoid confusion to readers, we reduced the use of binarization. We modified these as: takeaway food consumption/week (0, 1-2 servings or >2 servings), fruits consumption/d (<1 servings, 1 serving or >1 servings), dried fruits consumption/d (0, <1 servings or ≥1 servings) and so on.
In the Materials and Methods, sentences were supplemented and modified:
“information on demographic characteristics was collected through the self-administered questionnaire: gender, father’s education (below elementary school, elementary school, junior school, high school or technical secondary school, junior college or above), mother’s education (below elementary school, elementary school, junior school, high school or technical secondary school, junior college or above), race (han vs. minority), location (rural vs. urban), the only child (no vs. yes), self-rated family income (low, middle or high).
Likewise, living habits characteristics were assessed from a subjective way. For instance, “takeaway food consumption/week” was estimated from “In the last week, how many servings did you have takeaway food on average?” with response options of: “none”, “1-2”, “2-4” and “>4”. Subsequently, we merged the last two options as >2. Breakfast consumption (<7 days/week vs. 7 days/week), vegetables consumption/d (<3 servings vs. ≥3 servings), fruits consumption/d (<1 servings, 1 serving or >1 servings), dried fruits consumption/d (0, <1 servings or ≥1 servings), pure juice consumption (> 250ml)/d (0, <1 servings or ≥1 servings) and smoking use (no vs. yes) in the last week were assessed. Similarly, in the last month, soft drinks consumption (>250ml)/week (0, 1 serving and >1 servings), sugar-sweetened beverage consumption (> 250ml)/week (0, 1 serving and >1 servings), alcohol consumption (no vs. yes), duration of physical exercise each time (<60 min vs. ≥60 min), number of physical exercise (<1/d vs. ≥1/d), weekday screen time/d (<2h, <4h or ≥4h) and weekend screen time/d (<2h, <4h or ≥4h) were assessed.” (Page 4, lines173-193 marked in blue)
Table 2 was modified as:
Table 2. Living habits characteristics of the study population stratified by sex.
|
Variables |
All |
Males |
Females |
χ2 |
P |
|
Takeaway food consumption/week |
|
|
|
14.944 |
0.001 |
|
0 |
141(31.8) |
42(27.3) |
99(34.1) |
|
|
|
1-2 servings |
188(42.3) |
53(34.4) |
135(46.6) |
|
|
|
>2 servings |
115(25.9) |
59(38.3) |
56(19.3) |
|
|
|
Breakfast consumption/week |
|
|
|
5.369 |
0.020 |
|
<7 days |
276(62.2) |
107(69.5) |
169(58.3) |
|
|
|
7 days |
168(37.8) |
47(30.5) |
121(41.7) |
|
|
|
Vegetables consumption/d |
|
|
|
0.737 |
0.391 |
|
<3 servings |
342(77.0) |
115(74.7) |
227(78.3) |
|
|
|
≥3 servings |
102(23.0) |
39(25.3) |
63(21.7) |
|
|
|
Fruits consumption/d |
|
|
|
16.067 |
<0.001 |
|
<1 servings |
210(47.3) |
88(58.3) |
122(41.6) |
|
|
|
1 serving |
126(28.4) |
42(27.3) |
84(29.0) |
|
|
|
>1 servings |
108(24.3) |
24(15.6) |
84(29.0) |
|
|
|
Dried fruits consumption/d |
|
|
|
1.400 |
0.497 |
|
0 |
238(53.6) |
82(53.2) |
156(53.8) |
|
|
|
<1 servings |
140(31.5) |
45(29.2) |
95(32.8) |
|
|
|
≥1 servings |
66(15.3) |
27(17.5) |
39(13.4) |
|
|
|
Pure juice consumption (> 250ml)/d |
|
|
|
11.307 |
0.004 |
|
0 |
289(65.1) |
82(53.2) |
207(71.4) |
|
|
|
<1 servings |
115(25.9) |
48(31.2) |
67(23.1) |
|
|
|
≥1 servings |
40(9.0) |
24(15.6) |
16(75.5) |
|
|
|
Soft drinks consumption (> 250ml)/week |
|
|
|
15.286 |
<0.001 |
|
0 |
177(39.9) |
44(28.6) |
133(45.9) |
|
|
|
1 serving |
138(31.1) |
46(29.9) |
92(31.7) |
|
|
|
>1 servings |
129(29.1) |
64(41.6) |
65(22.4) |
|
|
|
Sugar-sweetened beverage consumption (> 250ml)/week |
|
|
|
3.129 |
0.209 |
|
0 |
111(25.0) |
38(24.7) |
73(25.2) |
|
|
|
1 serving |
140(31.5) |
44(28.6) |
96(33.1) |
|
|
|
>1 servings |
193(43.5) |
72(46.8) |
121(41.7) |
|
|
|
Alcohol consumption |
|
|
|
26.115 |
<0.001 |
|
No |
391(88.1) |
119(77.3) |
272(93.8) |
|
|
|
Yes |
53(11.9) |
35(22.7) |
18(6.2) |
|
|
|
Smoking |
|
|
|
7.798 |
0.005 |
|
No |
436(98.2) |
147(95.5) |
289(99.7) |
|
|
|
Yes |
8(1.8) |
7(4.5) |
1(0.3) |
|
|
|
Duration of physical exercise each time |
|
|
|
13.190 |
<0.001 |
|
<60 min |
411(92.6) |
133(86.4) |
278(95.9) |
|
|
|
≥60 min |
33(7.4) |
21(13.6) |
12(4.1) |
|
|
|
Number of physical exercise/d |
|
|
|
2.890 |
0.089 |
|
<1 |
371(83.6) |
135(87.7) |
236(81.4) |
|
|
|
≥1 |
73(16.4) |
19(12.3) |
54(18.6) |
|
|
|
Weekday screen time/d |
|
|
|
4.031 |
0.133 |
|
<2h |
121(27.3) |
32(20.8) |
89(30.7) |
|
|
|
<4h |
192(43.2) |
75(48.7) |
117(40.3) |
|
|
|
≥4h |
131(29.5) |
47(30.5) |
84(29.0) |
|
|
|
Weekend screen time/d |
|
|
|
15.653 |
0.016 |
|
<2h |
73(16.4) |
27(17.5) |
46(15.9) |
|
|
|
<4h |
159(35.5) |
50(32.5) |
109(37.6) |
|
|
|
≥4h |
212(47.7) |
77(50.0) |
135(46.6) |
|
|
Note: data are expressed as number (%).
Generally, in terms of living habits, takeaway food consumption/week, breakfast consumption, vegetables consumption/d, fruits consumption/d, dried fruits consumption/d, pure juice consumption (> 250ml)/d and smoking use were estimated by asking participants “In the last week, how many servings did you have”. Similarly, in the last month, soft drinks consumption (> 250ml)/week, sugar-sweetened beverage consumption (> 250ml)/week, alcohol consumption, duration of physical exercise each time, number of physical exercise, weekday screen time/d and weekend screen time/d were assessed.
Q12. Line 242-262 – please present this data in a table to make it easier for the readers.
Response: We thank the reviewer for his/her insightful suggestions and comments.
We presented this data in Table A7, Figure. 1. Probably due to our carelessness and mistakes in submitting the manuscript and related files, we failed to upload the supplementary materials. We feel sorry that the reviewer was not successful in seeing the Table A7.
We renamed Table A7 as Table 5, and supplemented it in the revised manuscript:
Table 5 Association of body composition parameters with different sleep characteristics of study population
|
Variables |
Visceral fat area(cm2) |
|
Body mass index(kg/m2) |
|
Waist to height ratio |
|
Waist to hip ratio |
||||||||||||
|
model 1 B(95%CI) |
model 1 B(95%CI) |
|
model 1 B(95%CI) |
model 1 B(95%CI) |
|
model 1 B(95%CI) |
model 1 B(95%CI) |
|
model 1 B(95%CI) |
model 1 B(95%CI) |
|||||||||
|
Weekday nap duration/d |
1.258 |
(-4.148,6.664) |
1.500 |
(-3.885,6.884) |
|
0.879 |
(0.119,1.640) |
0.678 |
(-0.102,1.458) |
|
0.011 |
(-<0.001,0.023) |
0.013 |
(0.001,0.025) |
|
0.011 |
(0.001,0.022) |
0.010 |
(-0.001,0.020) |
|
Weekend nap duration/d |
0.964 |
(-3.765,5.692) |
1.181 |
(-3.451,5.812) |
|
0.250 |
(-0.419,0.918) |
0.158 |
(-0.515,0.830) |
|
0.005 |
(-0.006,0.015) |
0.004 |
(-0.007,0.014) |
|
0.007 |
(-0.002,0.016) |
0.006 |
(-0.003,0.015) |
|
Average sleep duration |
7.338 |
(-0.170,14.846) |
6.963 |
(-0.227,14.152) |
|
1.172 |
(0.112, 2.232) |
1.117 |
(0.045,2.188) |
|
0.017 |
(<0.001,0.033) |
0.016 |
(<0.001,0.033) |
|
0.011 |
(-0.004,0.025) |
0.010 |
(-0.005,0.024) |
|
Social jetlag |
4.860 |
(-0.577,10.296) |
7.475 |
(2.137,12.813) |
|
0.838 |
(0.071, 1.605) |
0.878 |
(0.086,1.671) |
|
0.011 |
(-0.001, 0.023) |
0.015 |
(0.002,0.027) |
|
0.012 |
(0.002,0.022) |
0.012 |
(0.001,0.023) |
|
Average sleep efficiency/d |
0.386 |
(-0.167,0.940) |
0.449 |
(-0.088,0.987) |
|
0.035 |
(-0.044,0.113) |
0.038 |
(-0.040,0.116) |
|
0.001 |
(-0.001,0.002) |
0.001 |
(-<0.001,0.002) |
|
0.001 |
(-<0.001,0.002) |
0.001 |
(-<0.001,0.002) |
|
Screen time before sleep(> 0.5h) |
6.032 |
(-1.420,13.483) |
7.934 |
(0.700,15.167) |
|
0.568 |
(-0.487, 1.624) |
0.794 |
(-0.288,1.876) |
|
0.015 |
(-0.001,0.031) |
0.017 |
(0.001,0.033) |
|
0.015 |
(0.001,0.029) |
0.016 |
(0.002,0.030) |
continued
|
Variables |
Fat mass index |
|
Free fat mass index |
|
Body fat percentage(%) |
|||||||||
|
model 1 B(95%CI) |
model 1 B(95%CI) |
|
model 1 B(95%CI) |
model 1 B(95%CI) |
|
model 1 B(95%CI) |
model 1 B(95%CI) |
|||||||
|
Weekday nap duration/d |
0.267 |
(-0.327,0.861) |
0.196 |
(-0.364,0.757) |
|
0.673 |
(0.127,1.019) |
0.060 |
(-0.277,0.397) |
|
0.945 |
(-0.858,2.747) |
1.003 |
(-0.545,2.551) |
|
Weekend nap duration/d |
0.069 |
(-0.451,0.588) |
0.116 |
(-0.366,0.599) |
|
0.099 |
(-0.293,0.492) |
0.029 |
(-0.260,0.319) |
|
0.201 |
(-1.378,1.779) |
0.017 |
(-1.317,1.351) |
|
Average sleep duration |
0.747 |
(-0.084,1.577) |
0.743 |
(-0.018,1.503) |
|
0.452 |
(-0.173,1.078) |
0.395 |
(-0.072,0.862) |
|
1.632 |
(-0.878,4.142) |
1.655 |
(-0.223,3.534) |
|
Social jetlag |
0.289 |
(-0.313,0.892) |
0.663 |
(0.099,1.228) |
|
0.616 |
(0.166,1.066) |
0.297 |
(-0.046,0.640) |
|
0.094 |
(-1.725,1.914) |
1.703 |
(0.301,3.105) |
|
Average sleep efficiency/d |
0.045 |
(-0.016,0.105) |
0.052 |
(-0.003,0.108) |
|
0.011 |
(-0.035,0.057) |
0.152 |
(-0.076,0.380) |
|
0.025 |
(-0.160,0.210) |
0.051 |
(-0.104,0.206) |
|
Screen time before sleep(> 0.5h) |
0.734 |
(-0.090,1.557) |
0.902 |
(0.138,1.666) |
|
-0.193 |
(-0.814,0.428) |
-0.161 |
(-0.633,0.310) |
|
2.464 |
(-0.018,4.946) |
2.892 |
(1.014,4.771) |
model 1: A simple linear regression analysis of the sleep variables and parameters on body composition was conducted after no adjustment.
model 2: A multiple linear regression analysis of the sleep variables and parameters on body compositions was conducted after adjustment for gender, father’s education, mother’s education,
breakfast consumption, fruits consumption, pure juice consumption, soft drinks consumption, alcohol consumption, smoking, duration of physical exercise each time.
Q13. Line 315-316 – Please elaborate on that topic. Why is this observation important, are there any compounds present in vegetables related to obesity prevention?
Response: We thank the reviewer for his/her valuable comments and suggestions.
Individuals aged 18-25 years with high social jetlag tended to consume lower quantity of recommended amount of vegetables and fruits, which caused poor health outcomes including an increased risk of overweight and obesity (Zeron-Rugerio et al., 2019).Evidence suggests that bioactive compounds present in vegetables and fruits, containing carotenoids, polyphenols, and phytosterols, possibly have beneficial associations with the prevention of obesity and other diseases (Santos et al., 2022, Murillo-Castillo et al., 2020). This observation is of vital importance because it provides a different and novel idea for preventing overweight or obesity.
To elaborate the topic, in the Discussion, the sentence was modified as “Consuming lower quantity of recommended amount of vegetables and fruits increased the rise of overweight and obesity [57] as evidence suggests that bioactive compounds present in vegetables and fruits, containing carotenoids, polyphenols, and phytosterols, possibly have beneficial associations with the prevention of obesity and other diseases [58].” (Page 12, lines351-355 marked in blue)
The following reference was supplemented in the Reference List (marked in blue):
58.Santos DCD, Oliveira Filho JG, Sousa TL, Ribeiro CB, Egea MB: Ameliorating effects of metabolic syndrome with the consumption of rich-bioactive compounds fruits from Brazilian Cerrado: a narrative review. Crit Rev Food Sci Nutr 2022, 62(27):7632-7649.
References
- Obesity: preventing and managing the global epidemic. Report of a WHO consultation. World Health Organ Tech Rep Ser, 894, i-xii, 1-253.
ADAN, A., ARCHER, S. N., HIDALGO, M. P., DI MILIA, L., NATALE, V. & RANDLER, C. 2012. Circadian typology: a comprehensive review. Chronobiol Int, 29, 1153-75.
CATAROZOLI, C. 2023. Sleep During the Pandemic. Sleep Med Clin, 18, 219-224.
CHOI, G., YOON, H. J., SONG, Y. J., JEONG, H. M., GU, J. E., HAN, M., KIM, S. H., YOON, J. W. & KIM, H. 2022. Consistency of the estimated target weights and ECW/TBW using BIA after hemodialysis in patients between standing and lying-down positions. BMC Nephrol, 23, 106.
GAO, M., WEI, Y. X., LYU, J., YU, C. Q., GUO, Y., BIAN, Z., PEI, P., DU, H. D., CHEN, J. S., CHEN, Z. M., HUANG, T., LI, L. M. & CHINA KADOORIE BIOBANK COLLABORATIVE, G. 2019. [The cut-off points of body mass index and waist circumference for predicting metabolic risk factors in Chinese adults]. Zhonghua Liu Xing Bing Xue Za Zhi, 40, 1533-1540.
JAHRAMI, H. A., ALHAJ, O. A., HUMOOD, A. M., ALENEZI, A. F., FEKIH-ROMDHANE, F., ALRASHEED, M. M., SAIF, Z. Q., BRAGAZZI, N. L., PANDI-PERUMAL, S. R., BAHAMMAM, A. S. & VITIELLO, M. V. 2022. Sleep disturbances during the COVID-19 pandemic: A systematic review, meta-analysis, and meta-regression. Sleep Med Rev, 62, 101591.
LI, J. C., LYU, J., GAO, M., YU, C. Q., GUO, Y., BIAN, Z., PEI, P., DU, H. D., CHEN, J. S., CHEN, Z. M., LI, L. M. & CHINA KADOORIE BIOBANK COLLABORATIVE, G. 2019. [Association of body mass index and waist circumference with major chronic diseases in Chinese adults]. Zhonghua Liu Xing Bing Xue Za Zhi, 40, 1541-1547.
MCLESTER, C. N., NICKERSON, B. S., KLISZCZEWICZ, B. M. & MCLESTER, J. R. 2020. Reliability and Agreement of Various InBody Body Composition Analyzers as Compared to Dual-Energy X-Ray Absorptiometry in Healthy Men and Women. J Clin Densitom, 23, 443-450.
MONGRAIN, V., LAVOIE, S., SELMAOUI, B., PAQUET, J. & DUMONT, M. 2004. Phase relationships between sleep-wake cycle and underlying circadian rhythms in Morningness-Eveningness. J Biol Rhythms, 19, 248-57.
MONTARULI, A., CASTELLI, L., MULE, A., SCURATI, R., ESPOSITO, F., GALASSO, L. & ROVEDA, E. 2021. Biological Rhythm and Chronotype: New Perspectives in Health. Biomolecules, 11.
MURILLO-CASTILLO, K. D., FRONGILLO, E. A., CORELLA-MADUENO, M. A. & QUIZAN-PLATA, T. 2020. Food Insecurity Was Associated with Lower Fruits and Vegetables Consumption but Not with Overweight and Obesity in Children from Mexican Fishing Communities. Ecol Food Nutr, 59, 420-435.
NIE, X., CHEN, J., MA, X., NI, Y., SHEN, Y., YU, H., PANAGIOTOU, G. & BAO, Y. 2020. A metagenome-wide association study of gut microbiome and visceral fat accumulation. Comput Struct Biotechnol J,18, 2596-2609.
PAN, X. F., WANG, L. & PAN, A. 2021. Epidemiology and determinants of obesity in China. Lancet Diabetes Endocrinol, 9, 373-392.
SANTOS, D. C. D., OLIVEIRA FILHO, J. G., SOUSA, T. L., RIBEIRO, C. B. & EGEA, M. B. 2022. Ameliorating effects of metabolic syndrome with the consumption of rich-bioactive compounds fruits from Brazilian Cerrado: a narrative review. Crit Rev Food Sci Nutr, 62, 7632-7649.
WIECH, P., WOLOSZYN, F., TROJNAR, P., SKORKA, M. & BAZALINSKI, D. 2022. Does Body Position Influence Bioelectrical Impedance? An Observational Pilot Study. Int J Environ Res Public Health, 19.
ZERON-RUGERIO, M. F., CAMBRAS, T. & IZQUIERDO-PULIDO, M. 2019. Social Jet Lag Associates Negatively with the Adherence to the Mediterranean Diet and Body Mass Index among Young Adults. Nutrients, 11.

Reviewer 2 Report
The paper by Song and colleagues investigated the association between sleep behaviors and body compositions among Chinese college students. Overall, results show nap duration, sleep duration, social jetlag, screen time before sleep were linked to various measures of being overweight or obese. This manuscript would make a valuable contribution to current literature, however there are some concerns and remarks that would require attention. Please see comments below.
Abstract:
Line 25-26: sleep disorders were closely related to obesity and body composition indicators, should be rephrased as the study did not assess for specific sleep disorders but rather facets of sleep. Perhaps says something as such “Sleep characteristics were closely related to obesity and body composition indicators.”
Introduction:
Line 38-41- would recommend mentioning the geographic locations for the study mentioned involving college students.
Line 49: “Actually, we spend a third of our hours for sleep and it plays an essential part in our life.” Possibly to something like this, “we spend a third of our day sleeping as it plays an essential part in our life.”
Line 55-56- consider removing as the mentioned study is not relevant to the population of interest in this study. If choosing to keep in place, for line 57-please clarify what LM stands for, to coincide with the rest of the paragraph.
Line 63-65: consider removing as mentioned study is not relevant in age, to the population in this study.
Line 65-67- consider rephrasing sentence
Line 82-84- please add citations
Methods/Results:
1) Linear models were used despite acknowledging the data are non-normally distributed, violation of a core requirement of a linear model.
2) Analyses need to be adjusted for multiple comparisons (e.g., by the family-wise error rate) for each outcome - they tested 11 variables on each of 13 outcomes, which is 143 tests.
3) the reason the beta coefficients don't have units is they appear to have binarized all of the independent variables (alcohol? yes/no, physical exercise yes/no), which is lazy AF. Where possible, they should use the raw, unbinarized data and report beta coefficients with relevant units (e.g., 0.09 kg per hour of daytime napping or something like that).
Discussion:
Line 327-328: Please rephrase/recheck sentence.
Line 331-333: feels like depression was mentioned with no context. Please rephrase or consider removing all together.
Line 347-349- consider removing, not relevant to population of interest in this study. Lines 349-352 would suffice in getting point across.
Line 364-367- would be helpful to understand why the various confounders were chosen.
In limitations, may also be helpful to mention other potential confounders not accounted for such as medical conditions including underlying sleep disorder diagnoses.
Author Response
Dear Editors and Reviewers:
We would like to thank you for giving us a chance to revise our manuscript entitled “Sleep-body Composition Relation: Roles of Sleep Behaviors in General and Abdominal Obesity in Chinese College Students (ID: nutrients-2547197)” by Song et al. We also thank the reviewers for their valuable comments and suggestions to improve the quality of our manuscript. We have revised the manuscript accordingly. A point-by-point response to the reviewer’s comments are listed below.
Reviewer #2:
The paper by Song and colleagues investigated the association between sleep behaviors and body compositions among Chinese college students. Overall, results show nap duration, sleep duration, social jetlag, screen time before sleep were linked to various measures of being overweight or obese. This manuscript would make a valuable contribution to current literature, however there are some concerns and remarks that would require attention. Please see comments below.
Q1. Abstract: Line 25-26: sleep disorders were closely related to obesity and body composition indicators, should be rephrased as the study did not assess for specific sleep disorders but rather facets of sleep. Perhaps says something as such “Sleep characteristics were closely related to obesity and body composition indicators.”
Response: We agree with the reviewer’s constructive suggestions.
In the revised abstract, we modified the following sentence: “We found poor sleep characteristicswere closely related to general and abdominal obesity.” (Page 1, lines 23-24, marked in blue)
Q2. Introduction:
1). Line 38-41- would recommend mentioning the geographic locations for the study mentioned involving college students.
Response: We agree with the reviewer’s insightful and constructive suggestions.
Mentioning the geographical location of the mentioned college students can provide readers with a clearer understanding of the adverse impacts of weight gain particularly in fat mass.
In the Introduction, we modified the following sentence:
“A study involving college students recruited from campus in Beijing, China underlined the necessity of fat mass (FM) and fat free mass (FFM) control to avoid the risk of premature death.” (Page 16, lines 31-34, marked in blue)
2). Line 49: “Actually, we spend a third of our hours for sleep and it plays an essential part in our life.” Possibly to something like this, “we spend a third of our day sleeping as it plays an essential part in our life.”
Response: We express our gratitude to the reviewer for his/her useful suggestions.
In the Introduction, we modified the following sentence:
“We spend a third of our day sleeping as it plays an essential part in our life.” (Page 2, lines 61-62, marked in blue)
3). Line 55-56- consider removing as the mentioned study is not relevant to the population of interest in this study. If choosing to keep in place, for line 57-please clarify what LM stands for, to coincide with the rest of the paragraph.
Response: We agree with the reviewer’s useful and constructive suggestions.
In the Introduction, we removed the mentioned study.
4). Line 63-65: consider removing as mentioned study is not relevant in age, to the population in this study.
Response: We agree with the reviewer’s constructive suggestion and comment.
In the Introduction, we removed the mentioned study.
5). Line 65-67- consider rephrasing sentence
Response: We agree with the reviewer’s comment and suggestion.
In the Introduction, we rephased the sentence “However, another study containing 390 healthy adults (21-35 years) exhibited SJL had no influence on any obesity related anthropometric indices[26].” (Page 2, lines 80-82, marked in blue)
6). Line 82-84- please add citations
Response: We agree with the reviewer’s comment and suggestion.
In the Introduction, sentences were modified and supplemented:
“Overall, previous studies have supported a role of poor sleep in the development of obesity [30-32]. Literatures regarding sleep duration [33] or obstructive sleep apnea [34][35] merely used BMI and waist circumference (WC) to define general and abdominal obesity.” (Page 3, lines 91-93, marked in blue)
The following reference was supplemented in the Reference List (marked in blue):
- Jurado-Fasoli L, Amaro-Gahete FJ, De-la OA, Dote-Montero M, Gutierrez A, Castillo MJ: Association between Sleep Quality and Body Composition in Sedentary Middle-Aged Adults. Medicina (Kaunas) 2018, 54(5).
- Seo SH, Shim YS: Association of Sleep Duration with Obesity and Cardiometabolic Risk Factors in Children and Adolescents: A Population-Based Study. Sci Rep 2019, 9(1):9463.
- Mongrain V, Lavoie S, Selmaoui B, Paquet J, Dumont M: Phase relationships between sleep-wake cycle and underlying circadian rhythms in Morningness-Eveningness. J Biol Rhythms 2004, 19(3):248-257.
- Lopez-Garcia E, Faubel R, Leon-Munoz L, Zuluaga MC, Banegas JR, Rodriguez-Artalejo F: Sleep duration, general and abdominal obesity, and weight change among the older adult population of Spain. Am J Clin Nutr 2008, 87(2):310-316.
- Chen X, Pensuksan WC, Lohsoonthorn V, Lertmaharit S, Gelaye B, Williams MA: Obstructive Sleep Apnea and Multiple Anthropometric Indices of General Obesity and Abdominal Obesity among Young Adults. Int J Soc Sci Stud 2014, 2(3):89-99.
- Zhao X, Xu H, Qian Y, Liu Y, Zou J, Yi H, Guan J, Yin S: Abdominal Obesity Is More Strongly Correlated with Obstructive Sleep Apnea than General Obesity in China: Results from Two Separated Observational and Longitudinal Studies. Obes Surg 2019,29(8):2535-2547.
Q3. Methods/Results:
1). Linear models were used despite acknowledging the data are non-normally distributed, violation of a core requirement of a linear model.
Response: We agree with the reviewer’s comment and suggestion.
In our study, we applied Shapiro-Wilk method to examine normality for continuous data including several sleep variables and body composition parameters. The results showed that continuous variables were approximately normal distribution rather than complete. As we all known, there are 8 applicable conditions that need to be met before applying the multiple linear regression.
We summarized the conditions as follows:
(1) There is a linear relationship between the independent variable and the dependent variable.
(2) The residuals are independent of each other;
(3) The residual follows a normal distribution;
(4) Residual has homogeneity of variance;
(5) The dependent variable is a continuous variable;
(6) The independent variable is a continuous or categorical variable;
(7) There is no multicollinearity between independent variables;
(8) The sample size should be more than 20 times the independent variable.
We used partial regression plots to observe the linear relationship and Durbin-Watson values were approximately to 2. Besides, histograms and P-P plots showed an approximately normal distribution. Body composition indicators were continuous variables and sleep characteristics were continuous or categorical variables in our study. The current research included 444 university students and chose 10 confounders.
We apologize for the rough description in the manuscript previously. Instead of a simple description, we should elaborate the process to make it more detailed and rigorous.
2). Analyses need to be adjusted for multiple comparisons (e.g., by the family-wise error rate) for each outcome - they tested 11 variables on each of 13 outcomes, which is 143 tests.
Response: We agree with the reviewer’s comment and suggestion.
To make the analyses more accurate, we applied the Benjamini-Hochberg method to control the false discovery rate (FDR) for the multiple linear regression analyses with FDR-corrected P< 0.1 considered significant (Shuai et al., 2021).
In the Materials and Methods, the sentence was supplemented:
“The Benjamini-Hochberg method was used to control the false discovery rate (FDR) for the multiple linear regression analyses with FDR-corrected P< 0.1 considered significant [41].” (Page 5, lines 217-219, marked in blue)
The following reference was supplemented in the Reference List (marked in blue):
- Shuai M, Zuo LS, Miao Z, Gou W, Xu F, Jiang Z, Ling CW, Fu Y, Xiong F, Chen YM et al: Multi-omics analyses reveal relationships among dairy consumption, gut microbiota and cardiometabolic health. EBioMedicine 2021, 66:103284.
We modified the 3.4. Multiple linear regression analyses between sleep characteristics and body composition indicators of Results as:
“In multivariate linear regression analysis (Table 5, Figure. 1), after adjusting for gender, father’s education, mother’s education, breakfast consumption, fruits consumption, pure juice consumption (>250ml), soft drinks consumption (>250ml), alcohol consumption, smoking, duration of physical exercise each time, individuals with weekday nap duration/d (>30min) had higher WHtR (B=0.013, FDR-corrected P=0.080) (Figure 1c). Compared with the average sleep duration (>7h/d), participants with the average sleep duration (≤7h/d) showed a significant increase of 1.117 kg/m2 in BMI (FDR-corrected P=0.176) (Figure 1a) and 0.016 in WHtR (FDR-corrected P=0.080) (Figure 1c). There was also a trend for adolescents with great social jetlag(>1h) to have more VFA (B=7.475), WHtR (B=0.015), WHR (B=0.012), FMI (B=0.663) and BFP (B=1.703) (all FDR-corrected P <0.1). Spending more screen time before sleep (>0.5h) was associated with an increased risk for having higher VFA (B=7.934, FDR-corrected P=0.064) (Figure 1b), WHtR (B=0.017, FDR-corrected P=0.080) (Figure 1c), WHR (B=0.016, FDR-corrected P=0.090) (Figure 1c), FMI (B=0.902, FDR-corrected P=0.069) (Figure 1e) and BFP (B=2.892, FDR-corrected P=0.018)(Figure 1g). However, no other obvious associations were noted across sleep characteristics and body composition related indexes.” (Pages 8-9, lines 260-275, marked in blue)
Figure 1 was modified as:
Figure 1. Multiple linear regression analyses between sleep characteristics and body composition. Models were adjusted for gender, father’s education, mother’s education, breakfast consumption, fruits consumption, pure juice consumption, soft drinks consumption, alcohol consumption, smoking, duration of physical exercise each time. * Indicates that after the Benjamini-Hochberg false discovery rate (FDR) correction, the P value < 0.1.
3). the reason the beta coefficients don't have units is they appear to have binarized all of the independent variables (alcohol? yes/no, physical exercise yes/no), which is lazy AF. Where possible, they should use the raw, unbinarized data and report beta coefficients with relevant units (e.g., 0.09 kg per hour of daytime napping or something like that).
Response: We express our gratitude to the reviewer for his/her useful suggestions.
We feel guilty to have a majority of variables on living habits binarized.
In the Materials and Methods, sentences were modified as:
“Likewise, living habits characteristics were assessed from a subjective way. For instance, “takeaway food consumption/week” was estimated from “In the last week, how many servings did you have takeaway food on average?” with response options of: “none”, “1-2”, “2-4” and “>4”. Subsequently, we merged the last two options as >2. Breakfast consumption (<7 days/week vs. 7 days/week), vegetables consumption/d (<3 servings vs. ≥3 servings), fruits consumption/d (<1 servings, 1 serving or >1 servings), dried fruits consumption/d (0, <1 servings or ≥1 servings), pure juice consumption (> 250ml)/d (0, <1 servings or ≥1 servings) and smoking use (no vs. yes) in the last week were assessed. Similarly, in the last month, soft drinks consumption (> 250ml)/week (0, 1 serving and >1 servings), sugar-sweetened beverage consumption (> 250ml)/week (0, 1 serving and >1 servings), alcohol consumption (no vs. yes), duration of physical exercise each time (<60 min vs. ≥60 min), number of physical exercise (<1/d vs. ≥1/d), weekday screen time/d (<2h, <4h or ≥4h) and weekend screen time/d (<2h, <4h or ≥4h) were assessed.” (Pages 4, lines 180-193, marked in blue)
Table 2 was modified as:
Table 2. Living habits characteristics of the study population stratified by sex.
|
Variables |
All |
Males |
Females |
χ2 |
P |
|
Takeaway food consumption/week |
|
|
|
14.944 |
0.001 |
|
0 |
141(31.8) |
42(27.3) |
99(34.1) |
|
|
|
1-2 servings |
188(42.3) |
53(34.4) |
135(46.6) |
|
|
|
>2 servings |
115(25.9) |
59(38.3) |
56(19.3) |
|
|
|
Breakfast consumption/week |
|
|
|
5.369 |
0.020 |
|
<7 days |
276(62.2) |
107(69.5) |
169(58.3) |
|
|
|
7 days |
168(37.8) |
47(30.5) |
121(41.7) |
|
|
|
Vegetables consumption/d |
|
|
|
0.737 |
0.391 |
|
<3 servings |
342(77.0) |
115(74.7) |
227(78.3) |
|
|
|
≥3 servings |
102(23.0) |
39(25.3) |
63(21.7) |
|
|
|
Fruits consumption/d |
|
|
|
16.067 |
<0.001 |
|
<1 servings |
210(47.3) |
88(58.3) |
122(41.6) |
|
|
|
1 serving |
126(28.4) |
42(27.3) |
84(29.0) |
|
|
|
>1 servings |
108(24.3) |
24(15.6) |
84(29.0) |
|
|
|
Dried fruits consumption/d |
|
|
|
1.400 |
0.497 |
|
0 |
238(53.6) |
82(53.2) |
156(53.8) |
|
|
|
<1 servings |
140(31.5) |
45(29.2) |
95(32.8) |
|
|
|
≥1 servings |
66(15.3) |
27(17.5) |
39(13.4) |
|
|
|
Pure juice consumption (> 250ml)/d |
|
|
|
11.307 |
0.004 |
|
0 |
289(65.1) |
82(53.2) |
207(71.4) |
|
|
|
<1 servings |
115(25.9) |
48(31.2) |
67(23.1) |
|
|
|
≥1 servings |
40(9.0) |
24(15.6) |
16(75.5) |
|
|
|
Soft drinks consumption (> 250ml)/week |
|
|
|
15.286 |
<0.001 |
|
0 |
177(39.9) |
44(28.6) |
133(45.9) |
|
|
|
1 serving |
138(31.1) |
46(29.9) |
92(31.7) |
|
|
|
>1 servings |
129(29.1) |
64(41.6) |
65(22.4) |
|
|
|
Sugar-sweetened beverage consumption (> 250ml)/week |
|
|
|
3.129 |
0.209 |
|
0 |
111(25.0) |
38(24.7) |
73(25.2) |
|
|
|
1 serving |
140(31.5) |
44(28.6) |
96(33.1) |
|
|
|
>1 servings |
193(43.5) |
72(46.8) |
121(41.7) |
|
|
|
Alcohol consumption |
|
|
|
26.115 |
<0.001 |
|
No |
391(88.1) |
119(77.3) |
272(93.8) |
|
|
|
Yes |
53(11.9) |
35(22.7) |
18(6.2) |
|
|
|
Smoking |
|
|
|
7.798 |
0.005 |
|
No |
436(98.2) |
147(95.5) |
289(99.7) |
|
|
|
Yes |
8(1.8) |
7(4.5) |
1(0.3) |
|
|
|
Duration of physical exercise each time |
|
|
|
13.190 |
<0.001 |
|
<60 min |
411(92.6) |
133(86.4) |
278(95.9) |
|
|
|
≥60 min |
33(7.4) |
21(13.6) |
12(4.1) |
|
|
|
Number of physical exercise/d |
|
|
|
2.890 |
0.089 |
|
<1 |
371(83.6) |
135(87.7) |
236(81.4) |
|
|
|
≥1 |
73(16.4) |
19(12.3) |
54(18.6) |
|
|
|
Weekday screen time/d |
|
|
|
4.031 |
0.133 |
|
<2h |
121(27.3) |
32(20.8) |
89(30.7) |
|
|
|
<4h |
192(43.2) |
75(48.7) |
117(40.3) |
|
|
|
≥4h |
131(29.5) |
47(30.5) |
84(29.0) |
|
|
|
Weekend screen time/d |
|
|
|
15.653 |
0.016 |
|
<2h |
73(16.4) |
27(17.5) |
46(15.9) |
|
|
|
<4h |
159(35.5) |
50(32.5) |
109(37.6) |
|
|
|
≥4h |
212(47.7) |
77(50.0) |
135(46.6) |
|
|
Note: data are expressed as number (%).
Q4. Discussion:
1). Line 327-328: Please rephrase/recheck sentence.
Response: We express our gratitude to the reviewer for his/her suggestions.
However, we are sorry for failing to find the sentence maybe due to my inaccurate line number. We modified the whole discussion marked in blue.
2). Line 331-333: feels like depression was mentioned with no context. Please rephrase or consider removing all together.
Response: We express our gratitude to the reviewer for his/her useful suggestions.
In the Discussion, we removed the mentioned sentences.
3). Line 347-349- consider removing, not relevant to population of interest in this study.
Response: We express our gratitude to the reviewer for his/her useful suggestions.
In the Introduction, we removed the mentioned study.
4). Lines 349-352 would suffice in getting point across.
Response: We agree with the reviewer’s constructive suggestion and comment.
To avoid our lengthy explanation, we modified the sentences in the Discussion as:
“Studies regarding adolescents have demonstrated that screen time during bedtime has impacts on poor sleep outcomes (shorter sleep duration, longer sleep latency, lower sleep efficiency, etc.) [65,66], which can influence biological and social rhythms (e.g. evening chronotype, SJL) or circadian rhythm [64,67]. Interfering with excess screen time in bed and avoid sleep disorders, is a practical step to prevent obesity among adolescents effectively.” (Page 13, lines 375-381, marked in blue)
5). Line 364-367- would be helpful to understand why the various confounders were chosen.
Response: We express our gratitude to the reviewer for his/her comments.
We chose a variety of confounders from different dimensions to make the regression analysis more reasonable and to be accessible to more accurate results.
6). In limitations, may also be helpful to mention other potential confounders not accounted for such as medical conditions including underlying sleep disorder diagnoses.
Response: We thank the reviewer for his/her insightful and constructive suggestions.
We are sorry for our negligence at this point. When exploring the relationship between sleep characteristics and body composition in Chinese adolescents, medical conditions containing the diagnosis of potential sleep disorders are indeed nonnegligible confounders.
In the Discussion, the following sentence was supplemented:
“Finally, we didn’t consider genetic markers of obesity, possibility of familial obesity inheritance of participants, and factors or conditions related to sleep disturbance, including sleep apnea, insomniaand etc.” (Page 13, lines 411-414, marked in blue)
References
2022a. Older adolescent (15 to 19 years) and young adult (20 to 24 years) mortality.
2022b. Older children and young adolescent mortality (5 to 14 years). [Online]. Available: https://www.who.int/news-room/fact-sheets/detail/older-children-and-young-adolescent-mortality-(5-to-14-years) [Accessed].
AZZOPARDI, P. S., SAWYER, S. M., CARLIN, J. B., DEGENHARDT, L., BROWN, N., BROWN, A. D. & PATTON, G. C. 2018. Health and wellbeing of Indigenous adolescents in Australia: a systematic synthesis of population data. Lancet, 391, 766-782.
DOS SANTOS, R. R. G., FORTE, G. C., MUNDSTOCK, E., AMARAL, M. A., DA SILVEIRA, C. G., AMANTEA, F. C., VARIANI, J. F., BOOIJ, L. & MATTIELLO, R. 2020. Body composition parameters can better predict body size dissatisfaction than body mass index in children and adolescents. Eat Weight Disord, 25, 1197-1203.
JANKOVIC, N., SCHMITTING, S., KRUGER, B., NOTHLINGS, U., BUYKEN, A. & ALEXY, U. 2021. Changes in chronotype and social jetlag during adolescence and their association with concurrent changes in BMI-SDS and body composition, in the DONALD Study. Eur J Clin Nutr.
PFEFFERBAUM, A., KWON, D., BRUMBACK, T., THOMPSON, W. K., CUMMINS, K., TAPERT, S. F., BROWN, S. A., COLRAIN, I. M., BAKER, F. C., PROUTY, D., DE BELLIS, M. D., CLARK, D. B., NAGEL, B. J., CHU, W., PARK, S. H., POHL, K. M. & SULLIVAN, E. V. 2018. Altered Brain Developmental Trajectories in Adolescents After Initiating Drinking. Am J Psychiatry, 175, 370-380.
SHUAI, M., ZUO, L. S., MIAO, Z., GOU, W., XU, F., JIANG, Z., LING, C. W., FU, Y., XIONG, F., CHEN, Y. M. & ZHENG, J. S. 2021. Multi-omics analyses reveal relationships among dairy consumption, gut microbiota and cardiometabolic health. EBioMedicine, 66, 103284.
XIE, Y., WU, X., TAO, S., WAN, Y. & TAO, F. 2022. Development and validation of the self-rating of biological rhythm disorder for Chinese adolescents. Chronobiol Int, 39, 198-204.
ZHAO, Q., SULLIVAN, E. V., HONNORAT, N., ADELI, E., PODHAJSKY, S., DE BELLIS, M. D., VOYVODIC, J., NOONER, K. B., BAKER, F. C., COLRAIN, I. M., TAPERT, S. F., BROWN, S. A., THOMPSON, W. K., NAGEL, B. J., CLARK, D. B., PFEFFERBAUM, A. & POHL, K. M. 2021. Association of Heavy Drinking With Deviant Fiber Tract Development in Frontal Brain Systems in Adolescents. JAMA Psychiatry, 78, 407-415.

Reviewer 3 Report
Review Process for Nutrients
Mauscript ID: nutrients_2547197
Title: Sleep-body composition relation: roles of sleep behaviors in general and abdominal obesity in Chinese college students
Authors: Song, Gong, Lou, Zhou, Hao, Chen, Zhao, Jiang, Li, & Wang
This cross-sectional study investigated the relationship between sleep behaviours, defined in several ways (e.g., nap duration, average sleep duration, social jetlag, etc) and body composition, measured using bioelectrical impedance analysis, in 444 Chinese young adults. After controlling for several covariates, the main results showed that several sleep behaviours predicted body composition and obesity.
The paper is potentially interesting with a clear medical application. However, in my opinion, the paper is not suitable for the publication in the present form, and I would like to invite the authors to take into account my remarks to improve the quality of the study.
Title: I have honestly difficulty to consider participants involved in the study as a sample of young adults, given that the age range of participants was 17-22 years, that is a age range closer with the end of adolescents instead of adulthood. The mean age of the sample (about 19 years) confirmed my doubt. Thus, I suggest to change the title (and the other part of the paper) to focus more on adolescents instead of adulthood.
Introduction: In line with the previous point, the introduction should be more focused on the relationship between sleep and obesity/body composition in adolescent samples instead of a general relationship in the general population. This point, in my opinion, is important given that adolescents face to several changes at hormonal and behavioural levels, including sleep. A remarkable point for this particular “time-of-life” is related to chronotypes. In your study, you were far away from a “typical” distribution of chronotypes in the adults, whereas your distributions of morning-, intermediate-, and evening-types were more similar of chronotype distribution in adolescents (few MT and more ET). Also, the introduction is lacking of a novelty, a challenge, or a reliable reason to assess, for another time, the relationship between sleep and body composition. Probably, the novelty of the paper is related to bioelectrical impedance analysis but this part is less developed in the introduction and, particularly, it is unknown which additional information this analysis gave to the study and the literature.
Method: I have no particular problems with the subjective sleep reports and I really appreciated the attention adopted with the statistical analysis (non-normal distribution of the variables). My main concern here is related to the arbitrary choice to reduce a continuous variable in a categorical one. The average sleep latency is a variable in minutes and I did not understand how categorized it in below or above 30 minutes. In fact, this categorization is also problematic given that the international recommendation for a good sleep latency does not cover 30 minutes. In other words, a participant with an average sleep latency of 25 minutes/day has a sleep disorder in similar way to another participant with an average sleep latency of 40 minutes/day. The same for all other variables, especially for rMEQ score (after comparing the gender distribution of each chronotype, you can use the rMEQ score as it is). In addition, I hope that the authors had information for weekdays and weekend separately and, if it is so, I kindly recommend to keep these sleep variables separate. The sleep pattern of college students during the weekend is extremely different from that during weekdays. Thus, for example, the authors should calculate the average sleep duration during weekdays and that during weekends. The same for the midpoint of sleep, in addition to the social jetlag. These variables (as they are) should be inserted into the regression analyses. Finally, in the paper many statistical analyses were run with the same database (just for an example, 13 multiple linear regression analyses) and the risk of I type error increased exponentially. Thus, I recommend to either the application of p corrections or to fix the p value at very low threshold (e.g., p = .0001). As last point, the authors should perform logistic regression analysis to assess which sleep variables predict the critical body composition values. For instance, the authors had an objective measure of the BMI and in this age range, for both males and females, a BMI >= 30 should indicate obesity. The authors should test which sleep variables predict the probability to report a BMI>=30.
Results: this section seems to be clear but, unfortunately, I did not find tables A1-A7 neither in the supplementary materials (which is in the original version of manuscript template), neither in the appendices. Thus, I cannot say whether what it is mentioned in the Tables is well-written and useful for the paper. More importantly, I cannot say whether the discussion is data-driven or speculative.
Discussion: this part is affected by a lack of structure of the introduction. As in the introduction, the results were discussed with reference to adolescents, children, or adults but this way to discuss the data is confusing given that your sample is poorly comparable with other samples in the literature. A specific reference to a particular period of the life (adolescents for me but you can choose young adults if you will be able to create a convincible motivations) should be helpful for the discussion. At the same time, I encourage the authors to pay more attention to their conclusions. At page 10, lines 317-318, they wrote: “Nevertheless, the present study did not find notable associations between chronotype and obesity”. The social jetlag is considered a reliable measure of chronotypes given that it is sensible to identify extreme chronotypes with people with shorter or larger social jetlag. For this reason, the recommendation to include midpoint of sleep during weekdays and weekend, as well as the rMEQ score in your analysis should address better this aspect.
English mother-tongue is necessary
Author Response
Dear Editors and Reviewers:
We would like to thank you for giving us a chance to revise our manuscript entitled “Sleep-body Composition Relation: Roles of Sleep Behaviors in General and Abdominal Obesity in Chinese College Students (ID: nutrients-2547197)” by Song et al. We also thank the reviewers for their valuable comments and suggestions to improve the quality of our manuscript. We have revised the manuscript accordingly. A point-by-point response to the reviewer’s comments are listed below
Reviewer #3:
This cross-sectional study investigated the relationship between sleep behaviours, defined in several ways (e.g., nap duration, average sleep duration, social jetlag, etc) and body composition, measured using bioelectrical impedance analysis, in 444 Chinese young adults. After controlling for several covariates, the main results showed that several sleep behaviours predicted body composition and obesity.
The paper is potentially interesting with a clear medical application. However, in my opinion, the paper is not suitable for the publication in the present form, and I would like to invite the authors to take into account my remarks to improve the quality of the study.
Q1. Title: I have honestly difficulty to consider participants involved in the study as a sample of young adults, given that the age range of participants was 17-22 years, that is a age range closer with the end of adolescents instead of adulthood. The mean age of the sample (about 19 years) confirmed my doubt. Thus, I suggest to change the title (and the other part of the paper) to focus more on adolescents instead of adulthood.
Response: We agree with the reviewer’s insightful and constructive suggestions and comments.
The World Health Organization (WHO) defines adolescence as 10-19 years and young adults as 20-24 years (2022b, 2022a). Although a study conducted in Anhui Medical University (Hefei, Anhui, China) called the research subject young people with average 18.8 ± 0.9 years (freshmen or sophomores), we believe that it is more reasonable to consider participants as adolescents. The reasons are as follows.
In our study, the age range of the subjects was 17-22 years, with an average age of 19.12±1.177 years, which was more in line with the age range of adolescents. Besides, in many studies (Azzopardi et al., 2018, Pfefferbaum et al., 2018, Zhao et al., 2021), the age range of adolescence is considered prolonged such as 10-24 years (Azzopardi et al., 2018). What’s more, the questionnaire for Chinese adolescents applied in our study also extended subjects to university students (Xie et al., 2022). We prolonged this definition of adolescents to be consistent with the questionnaire development team as well.
In the revised manuscript, in order to perfect the title more accurate, we modified it as “Sleep-body Composition Relationship: Roles of Sleep Behaviors in General and Abdominal Obesity in Chinese Adolescents aged 17-22 years”.
In addition, we revised the inappropriate descriptions of our study samples in the whole manuscript.
In the Abstract:
“This study aimed to investigate the association between sleep behaviors and body composition, which was measured by bioelectrical impedance analysis (BIA) among Chinese adolescents.” (Page 1, lines 10-12, marked in blue)
In the Introduction:
“The present study aimed to evaluate the relationship between sleep characteristics and body composition in Chinese adolescents aged 17-22 years.” (Page 3, lines 102-103, marked in blue)
In the Discussion:
“Adolescents with average sleep duration (≤7h/d) showed obvious increases in BMI and WHtR. High SJL could positively affect general and abdominal obesity (observed by VFA, BMI, WHtR, WHR, FMI and BFP).” (Page 11, lines 300-302, marked in blue)
In the Conclusion:
“In this study, we showed that poor sleep characteristics got involved in the development of general and abdominal obesity among Chinses adolescents.” (Page 13, lines 415-417, marked in blue)
Nonetheless, we should be cautious about adolescent age range controversies, especially when the definitions of adolescents in the cited literatures are inconsistent with ours. To prevent confusing readers, we also showed age ranges when describing the participants.
In the Introduction:
“A study of adolescents (45% females, 9-17 years) emphasized SJL was related to obvious changes in fat mass index (FMI) (Jankovic et al., 2021).” (Page 2, lines 79-80, marked in blue)
In the Discussion:
“Among 534 young adults (18–25 years), individuals with greater SJL were related to a fewer intake of fruits and vegetables, even skipping breakfast [56].”
(Page 12, lines 349-351, marked in blue)
Q2. Introduction: In line with the previous point, the introduction should be more focused on the relationship between sleep and obesity/body composition in adolescent samples instead of a general relationship in the general population. This point, in my opinion, is important given that adolescents face to several changes at hormonal and behavioural levels, including sleep. A remarkable point for this particular “time-of-life” is related to chronotypes. In your study, you were far away from a “typical” distribution of chronotypes in the adults, whereas your distributions of morning-, intermediate-, and evening-types were more similar of chronotype distribution in adolescents (few MT and more ET).
Also, the introduction is lacking of a novelty, a challenge, or a reliable reason to assess, for another time, the relationship between sleep and body composition. Probably, the novelty of the paper is related to bioelectrical impedance analysis but this part is less developed in the introduction and, particularly, it is unknown which additional information this analysis gave to the study and the literature.
Response: We agree with the reviewer’s insightful and constructive suggestions and comments.
It is of vital importance to focus on the relationship between sleep and body composition rather than other variables. Moreover, we should concentrate on our own population and we have improved this point in the revised manuscript. We also took several measures to add our novelty in the Introduction such as the application of bioelectrical impedance analysis (BIA), indicators for assessing general and abdominal obesity, weekday-to-weekend sleep differences, etc.
We modified the Introduction:
“The prevalence of overweight and obesity has been increasingly serious during the last four decades, which has undoubtedly become a major public health concern in China. The national survey in 2018 demonstrated that the obesity rate of Chinses adolescents reached 16.0% [1,2].Abdominal obesity is defined as excess fat distribution in the abdominal area [3]. Waist-to-height ratio (WHtR), a maker of central adiposity, has strong associations with diabetes, hypertension [4] and cardiometabolic health [5]. A study involving college students recruited from campus in Beijing, China underlined the necessity of fat mass (FM) and fat free mass (FFM) control to avoid the risk of premature death. Abundant FM and low FFM represent general fat accumulation, which have adverse health effects especially in adolescents [6]. Compared to normal-weight peers, adolescents classified as obese have higher rates of morbidity and mortality, and higher risks of developing obesity as adults [1]. More attention should be paid to the population in establishing public health strategies. The novel coronavirus disease 2019 (COVID-19) has spread rapidly throughout countries. Available evidence suggested that adolescents with obesity were more inclined to develop severe medical conditions from COVID-19 [7,8]. Therefore, to confront the COVID-19 pandemic, it is of great significance to focus on general and abdominal obesity in adolescents.
Body composition is considered as an important predictor in various clinical scenarios including general and abdominal obesity. Body mass index (BMI) is a widely used indicator for defining obesity [9,10]. Based on Chinese criteria, overweight is defined as a BMI of 24.0 kg/m² and obesity as a BMI of 28.0 kg/m² or higher in adults (≥18 years) [11,12]. Nonetheless, BMI does not take into account the distribution of adipose tissue, and cannot distinguish between FM and FFM (Dos Santos et al., 2020), which in several situations is the key factor influencing disease risk. For instance, visceral adipose tissue is known to be more closely linked to cardiovascular risk than subcutaneous adipose tissue, thus it is more beneficial to assess waist circumference (WC) or waist to hip ratio (WHR) instead of BMI [13]. WC, WHtR and WHR ae key markers for identifying abdominal obesity [14]. Bioelectrical impedance analysis (BIA) and dual energy X-ray absorptiometry (DXA) are extensively used settings to evaluate body composition mainly in epidemiological and clinical analysis, respectively. DXA has better accuracy compared to BIA, but the safety of repeated measurements is worth improving [15].
The causes for excessive fat accumulation and high rise of obesity are complicated with biological, economic, social and cultural factors, and negative effects of poor sleep may be part of the determinants [16]. We spend a third of our day sleeping as it plays an essential part in our life [17]. A population-based longitudinal study with 1024 participants has shown that insufficient sleep leads to reduced leptin and elevated ghrelin levels, which can increase appetite thus promoting weight gain [18]. Of note, the COVID-19 pandemic has exacerbated the prevalence of sleep problems, which may contribute to shorter sleep duration, poor sleep quality, low sleep efficiency, etc. in all populations, especially the groups of children and adolescents [19,20]. Insufficient sleep duration and low sleep efficiency were associated with digestive disorders or conversely represented specific clinical manifestation of gastrointestinal diseases [21]. Studies conducted in China, Spain, the USA, and the UK demonstrated a significant increase in obesity occurrence when nap duration lasted longer than an hour [22]. It seems that individual's daytime napping is not always constant and may vary between weekdays and weekends. Early school start time and heavy study burden may contribute to student’s fatigue so that a short break is needed.
Social jetlag (SJL) is a dissonance between the biological clock and the social clock, which stems from students’ desire to make up for sleep loss during the weekdays by waking up later on weekends [23], especially in E-types, who have a later bedtime/wake-up time, and reach their best achievements during the second half of the day [24]. A study of adolescents (45% females, 9-17 years) emphasized SJL was related to obvious changes in fat mass index (FMI) [25]. However, another study containing 390 healthy adults (21-35 years) exhibited SJL had no influence on any obesity related anthropometric indices [26].
Owing to growing autonomy or changes in learning methods, college students tend to have an increase in screen media exposure [27], consequently with a high risk of obesity [28]. A cross-sectional study with 49,051 college and university students showed that screen use in bed had a strong negative association with sleep latency, sleep duration and sleep efficiency [29]. Therefore, due to adolescents’ preferred and unique pattern of biological rhythm, it is of great significance to understand how poor sleep characteristics yield impacts on obesity and body composition for the purpose of taking preventive measures and designing appreciated strategies.
Overall, previous studies have supported a role of poor sleep in the development of obesity [30-32].Literatures regarding sleep duration [33] or obstructive sleep apnea [34,35] merely used BMI and WC to define general and abdominal obesity. A majority of researchers have limited parameters in identifying general and central obesity. The present study took FMI, fat free mass index (FFMI),BMI, BFP, WHR and WHtR into account using Inbody 720, an analyzer utilizing a tetrapolar 8-point tactile electrode system and hand-to-foot BIA that sends varying frequencies of alternating current through the body, to make an accurate and generalized estimation for obesity. As adolescents are in a period of rapid change [36], it is necessary to focus on the unique population. Of note, there is a gap in the research about roles of poor sleep in the body composition in adolescents (aged 17–22 years), especially in China, the world’s most populous nation.
The present study aimed to evaluate the relationship between sleep characteristics and body composition in Chinese adolescents aged 17-22 years. Not only did we observe the various novel sleep variables but also we paid attention to the weekday-to-weekend sleep differences such as SJL in this special population. Additionally, we examined sex difference in sleep characteristics together with body composition parameters and adjusted for gender in linear regression analyses. Notably, the definition of adolescence was prolonged in the current study.”
The following reference was supplemented in the Reference List (marked in blue):
- Meng C, Yucheng T, Shu L, Yu Z: Effects of school-based high-intensity interval training on body composition, cardiorespiratory fitness and cardiometabolic markers in adolescent boys with obesity: a randomized controlled trial. BMC Pediatr 2022, 22(1):112.
- Sun M, Feng W, Wang F, Li P, Li Z, Li M, Tse G, Vlaanderen J, Vermeulen R, Tse LA: Meta-analysis on shift work and risks of specific obesity types. Obes Rev 2018, 19(1):28-40.
- Gotzinger F, Santiago-Garcia B, Noguera-Julian A, Lanaspa M, Lancella L, Calo Carducci FI, Gabrovska N, Velizarova S, Prunk P, Osterman V et al: COVID-19 in children and adolescents in Europe: a multinational, multicentre cohort study. Lancet Child Adolesc Health 2020, 4(9):653-661.
- Nogueira-de-Almeida CA, Del Ciampo LA, Ferraz IS, Del Ciampo IRL, Contini AA, Ued FDV: COVID-19 and obesity in childhood and adolescence: a clinical review. J Pediatr (Rio J) 2020, 96(5):546-558.
- Dos Santos RRG, Forte GC, Mundstock E, Amaral MA, da Silveira CG, Amantea FC, Variani JF, Booij L, Mattiello R: Body composition parameters can better predict body size dissatisfaction than body mass index in children and adolescents. Eat Weight Disord 2020, 25(5):1197-1203.
- Xu YX, Zhang AH, Yu Y, Wan YH, Tao FB, Sun Y: Sex-specific association of exposure to bedroom light at night with general and abdominal adiposity in young adults. Ecotoxicol Environ Saf 2021, 223:112561.
- Gao M, Wei YX, Lyu J, Yu CQ, Guo Y, Bian Z, Pei P, Du HD, Chen JS, Chen ZM et al: [The cut-off points of body mass index and waist circumference for predicting metabolic risk factors in Chinese adults]. Zhonghua Liu Xing Bing Xue Za Zhi 2019, 40(12):1533-1540.
- Pan XF, Wang L, Pan A: Epidemiology and determinants of obesity in China. Lancet Diabetes Endocrinol 2021, 9(6):373-392.
- Stich FM, Huwiler S, D'Hulst G, Lustenberger C: The Potential Role of Sleep in Promoting a Healthy Body Composition: Underlying Mechanisms Determining Muscle, Fat, and Bone Mass and Their Association with Sleep. Neuroendocrinology 2022, 112(7):673-701.
- Bojanic D, Ljubojevic M, Krivokapic D, Gontarev S: Waist circumference, waist-to-hip ratio, and waist-to-height ratio reference percentiles for abdominal obesity among Macedonian adolescents. Nutr Hosp 2020, 37(4):786-793.
- Marra M, Sammarco R, De Lorenzo A, Iellamo F, Siervo M, Pietrobelli A, Donini LM, Santarpia L, Cataldi M, Pasanisi F et al: Assessment of Body Composition in Health and Disease Using Bioelectrical Impedance Analysis (BIA) and Dual Energy X-Ray Absorptiometry (DXA): A Critical Overview. Contrast Media Mol Imaging 2019, 2019:3548284.
- Catarozoli C: Sleep During the Pandemic. Sleep Med Clin 2023, 18(2):219-224.
- Jahrami HA, Alhaj OA, Humood AM, Alenezi AF, Fekih-Romdhane F, AlRasheed MM, Saif ZQ, Bragazzi NL, Pandi-Perumal SR, BaHammam AS et al: Sleep disturbances during the COVID-19 pandemic: A systematic review, meta-analysis, and meta-regression. Sleep Med Rev 2022, 62:101591.
- Vernia F, Di Ruscio M, Ciccone A, Viscido A, Frieri G, Stefanelli G, Latella G: Sleep disorders related to nutrition and digestive diseases: a neglected clinical condition. Int J Med Sci 2021, 18(3):593-603.
- Cai Z, Yang Y, Zhang J, Liu Y: The relationship between daytime napping and obesity: a systematic review and meta-analysis. Sci Rep 2023, 13(1):12124.
- Jurado-Fasoli L, Amaro-Gahete FJ, De-la OA, Dote-Montero M, Gutierrez A, Castillo MJ: Association between Sleep Quality and Body Composition in Sedentary Middle-Aged Adults. Medicina (Kaunas) 2018, 54(5).
- Seo SH, Shim YS: Association of Sleep Duration with Obesity and Cardiometabolic Risk Factors in Children and Adolescents: A Population-Based Study. Sci Rep 2019, 9(1):9463.
- Mongrain V, Lavoie S, Selmaoui B, Paquet J, Dumont M: Phase relationships between sleep-wake cycle and underlying circadian rhythms in Morningness-Eveningness. J Biol Rhythms 2004, 19(3):248-257.
- Lopez-Garcia E, Faubel R, Leon-Munoz L, Zuluaga MC, Banegas JR, Rodriguez-Artalejo F: Sleep duration, general and abdominal obesity, and weight change among the older adult population of Spain. Am J Clin Nutr 2008, 87(2):310-316.
- Chen X, Pensuksan WC, Lohsoonthorn V, Lertmaharit S, Gelaye B, Williams MA: Obstructive Sleep Apnea and Multiple Anthropometric Indices of General Obesity and Abdominal Obesity among Young Adults. Int J Soc Sci Stud 2014, 2(3):89-99.
- Zhao X, Xu H, Qian Y, Liu Y, Zou J, Yi H, Guan J, Yin S: Abdominal Obesity Is More Strongly Correlated with Obstructive Sleep Apnea than General Obesity in China: Results from Two Separated Observational and Longitudinal Studies. Obes Surg 2019, 29(8):2535-2547.
- Kail R: Developmental change in speed of processing during childhood and adolescence. Psychol Bull 1991, 109(3):490-501.
Q3. Method: I have no particular problems with the subjective sleep reports and I really appreciated the attention adopted with the statistical analysis (non-normal distribution of the variables). My main concern here is related to the arbitrary choice to reduce a continuous variable in a categorical one. The average sleep latency is a variable in minutes and I did not understand how categorized it in below or above 30 minutes. In fact, this categorization is also problematic given that the international recommendation for a good sleep latency does not cover 30 minutes. In other words, a participant with an average sleep latency of 25 minutes/day has a sleep disorder in similar way to another participant with an average sleep latency of 40 minutes/day. The same for all other variables, especially for rMEQ score (after comparing the gender distribution of each chronotype, you can use the rMEQ score as it is). In addition, I hope that the authors had information for weekdays and weekend separately and, if it is so, I kindly recommend to keep these sleep variables separate. Thesleep pattern of college students during the weekend is extremely different from that during weekdays. Thus, for example, the authors should calculate the average sleep duration during weekdays and that during weekends. The same for the midpoint of sleep, in addition to the social jetlag. These variables (as they are) should be inserted into the regression analyses.
Finally, in the paper many statistical analyses were run with the same database (just for an example, 13 multiple linear regression analyses) and the risk of I type error increased exponentially. Thus, I recommend to either the application of p corrections or to fix the p value at very low threshold (e.g., p = .0001).
As last point, the authors should perform logistic regression analysis to assess which sleep variables predict the critical body composition values. For instance, the authors had an objective measure of the BMI and in this age range, for both males and females, a BMI >= 30 should indicate obesity. The authors should test which sleep variables predict the probability to report a BMI>=30.
Response: We appreciate your valuable advice
In the revised manuscript, we made changes to the sleep variables. We added weekday nap duration and weekend nap duration, removed several sleep characteristics, also added continuous variables. Moreover, to make the analyses more accurate, we applied the Benjamini-Hochberg method to control the false discovery rate (FDR) for the multiple linear regression analyses with FDR-corrected P<0.1 considered significant (Shuai et al., 2021). Besides, we had attempted logistic regression analysis but it didn't work well without good prediction effect.
Table 3 was modified:
Table 3. Sleep variables of the study population stratified by sex
|
Variables |
All |
Males |
Females |
χ2/t |
P |
|
Weekday nap duration/d |
|
|
|
13.852 |
<0.001 |
|
≤30min |
330(74.3) |
96(63.6) |
234(79.9) |
|
|
|
>30min |
114(25.7) |
55(36.4) |
59(20.1) |
|
|
|
Weekend nap duration/d |
|
|
|
0.338 |
0.561 |
|
≤30min |
232(52.3) |
76(50.3) |
156(53.2) |
|
|
|
>30min |
212(47.7) |
75(49.7) |
137(46.8) |
|
|
|
Average sleep duration/d |
|
|
|
0.102 |
0.749 |
|
≤7h |
49(11.0) |
18(11.7) |
31(10.7) |
|
|
|
>7h |
395(89.0) |
136(88.3) |
259(89.3) |
|
|
|
Social jetlag |
|
|
|
5.846 |
0.016 |
|
≤1h |
333(75.0) |
105(68.2) |
228(78.6) |
|
|
|
>1h |
111(25.0) |
49(31.8) |
62(21.4) |
|
|
|
Average sleep efficiency (%)/d |
95.2±4.3 |
95.2±4.7 |
95.3±4.0 |
0.335 |
0.738 |
|
Screen time before sleep (>0.5h) |
|
|
|
0.043 |
0.836 |
|
No |
50(11.3) |
18(11.7) |
32(11.0) |
|
|
|
Yes |
394(88.7) |
136(88.3) |
258(89.0) |
|
|
In the Materials and Methods,
2.3. Sleep characteristics was modified as:
“Weekday and weekend nap duration were obtained from “In the last month, how many minutes did you nap on average ?” with response options of ≤30 min/d or >30 min/d. Subjects were also asked to recall the questions (in the last month), “What time did you go to bed on average at night on weekdays and weekends?”; “How many minutes on average did you fall asleep after going to bed on weekdays and weekends?” and “What time did you get up on average in the morning on weekdays and weekends?”. Average sleep latency was calculated as: (weekday sleep latency × 5 + weekend sleep latency × 2)/7. Sleep duration was defined as: average time to wake up - average time to bed - average sleep latency and was divided into ≤7h/d and >7h/d [37]. Similarly, average sleep duration was calculated as: (weekday sleep duration × 5 + weekend sleep duration × 2)/7. Average sleep efficiency was defined as: a ratio of total sleep time to time in bed (×100%) [38] and a higher percentage indicated better sleep quality. Screen time before sleep (>0.5h) (no vs. yes) was estimated by the following question “Was your screen time before sleep more than 0.5h at night in the last month?”. SJL was defined as the difference in hours between the midpoints of sleep on weekdays (school days) versus weekends (free days) [23]. This was calculated by subtracting the midpoint of sleep on weekdays from that of weekends [39] and was divided into ≤1h and >1h [40] . ” (Pages 3-4, lines 145-162, marked in blue)
The sentences were supplemented:
“The Benjamini-Hochberg method was used to control the false discovery rate (FDR) for the multiple linear regression analyses with FDR-corrected P< 0.1 considered significant [41].” (Pages 5, lines 217-219, marked in blue)
The following reference was supplemented in the Reference List (marked in blue):
- Healthy China Action (2019-2030). 2019.
- Shuai M, Zuo LS, Miao Z, Gou W, Xu F, Jiang Z, Ling CW, Fu Y, Xiong F, Chen YM et al: Multi-omics analyses reveal relationships among dairy consumption, gut microbiota and cardiometabolic health. EBioMedicine 2021, 66:103284.
Q4. Results: this section seems to be clear but, unfortunately, I did not find tables A1-A7 neither in the supplementary materials (which is in the original version of manuscript template), neither in the appendices. Thus, I cannot say whether what it is mentioned in the Tables is well-written and useful for the paper. More importantly, I cannot say whether the discussion is data-driven or speculative.
Response: We thank the reviewer for his/her insightful suggestions and comments. Tables A1-A6aimed to determine confounding factors, using unpaired t-test and univariate one-way ANOVAsbetween the dependent variables and possible confounders (e.g. demographic characteristics and living habits characteristics). Table A7 was simple and multiple linear regression analysis and we renamed it as Table 5. Probably due to our carelessness and mistakes in submitting the manuscript and related files, we failed to upload the supplementary materials successfully. We feel sorry that the reviewer was not successful in seeing the Table A1-A7.
Q5. Discussion: this part is affected by a lack of structure of the introduction. As in the introduction, the results were discussed with reference to adolescents, children, or adults but this way to discuss the data is confusing given that your sample is poorly comparable with other samples in the literature. A specific reference to a particular period of the life (adolescents for me but you can choose young adults if you will be able to create a convincible motivations) should be helpful for the discussion.
At the same time, I encourage the authors to pay more attention to their conclusions. At page 10, lines 317-318, they wrote: “Nevertheless, the present study did not find notable associations between chronotype and obesity”. The social jetlag is considered a reliable measure of chronotypes given that it is sensible to identify extreme chronotypes with people with shorter or larger social jetlag. For this reason, the recommendation to include midpoint of sleep during weekdays and weekend, as well as the rMEQ score in your analysis should address better this aspect.
Response: We are thankful to the review for his/her constructive suggestions and comments.
We paid great attention to the subject of the cited literatures in the revised manuscript. To make the analysis more accurate, we adjusted sleep variables (e.g. removing some sleep variables and adding weekday nap duration and weekend nap duration)
Discussion was modified:
“In the current study, a significant association was observed between poor sleep behaviors and general and abdominal obesity. After adjusting for covariates, weekday nap duration (>30min/d) was positively related to higher WHtR values. Adolescents with average sleep duration (≤7h/d) showed obvious increases in BMI and WHtR. High SJL could positively affect general and abdominal obesity (observed by VFA, BMI, WHtR, WHR, FMI and BFP). More screen time before sleep (>0.5h) also exhibited higher risk of obesity by VFA, WHtR, WHR, FMI and BFP. To the best of our knowledge, this is the first cross-sectional study to investigate the association between various representative sleep characteristics with comprehensive body composition indicators measured by BIA in Chinese adolescents.
Our findings regarding average sleep duration were consistent with some previous literatures. For instance, a systematic research suggested that short sleep duration was associated with the risk of developing overweight or obesity among Chinese children and adolescents [42]. A study in China with 9059 participants (63.08% were females) revealed that short sleep duration caused the elevated rates of general obesity (defined using BMI) and visceral obesity with sex differences [43] The results acquired in Mexican children/adolescents exposed that the combined effects of inactivity, excessive screen time and inadequate sleep led to overweight/obesity and higher values of FM [44]. Several mechanisms have been proposed. Hormonal changes are thought to be involved in the relationship between sleep duration and obesity. Leptin is able to suppress appetite, while ghrelin has opposing functions of stimulating hunger [44]. A clinical research including 12 healthy young men demonstrated that when sleep time was restricted to 4 hours sleeping in bed at night over two days, subjects’ leptin level decreased by 18% and ghrelin level increased by 28% compared with hormonal levels of those spending 10 hours in bed [45]. However, some studies also demonstrated that between sleep restriction and normal sleep, the former had no significant effect on ghrelin and leptin levels [46,47]. While the underlying explanations need to be further explored, it is clear that short sleep duration brings daytime sleepiness and fatigue, which may result in a low physical activity [47]. Most likely, no separate factor explains the impact of short sleep on obesity completely, but multiple mechanisms work together. Conversely, obstructive sleep apnea frequently occurs among obese patients [48] and patients can have an improvement with weight loss [49]. Therefore, bidirectional relationships may link insufficient sleep duration to obesity [50,51].
Our results supported that high SJL had a positive impact on both general and abdominal obesity, which were in line with several previous literatures. Serving as a measure of circadian misalignment, SJL has played a role as a risk factor in obesity-related chronic diseases [51] and was significantly associated with an increased WC in a Japanese working population [52]. The explanatory mechanisms remain unclear. Some hypothesized that altered sleep-wake cycles could influence obesogenic environments of adipocytes and central circadian clock by modifying gene expression [39,53]. Emerging evidence indicates that circadian misalignment leads to increased weight and the development of obesity [54]. Apart from this, high SJL has associations with unhealthy dietary patterns, which are likely to cause overweight or obese. According to a study of 3060 adolescents in the United States, high SJL led to more choices to sweetened beverages and fast food, which caused obesity (defined using BMI) and other negative health outcomes [55]. Among 534 young adults (18–25 years), individuals with greater SJL were related to a fewer intake of fruits and vegetables, even skipping breakfast [56]. Consuming lower quantity of recommended amount of vegetables and fruits increased the rise of overweight and obesity [57] as evidence suggests that bioactive compounds present in vegetables and fruits, containing carotenoids, polyphenols, and phytosterols, possibly have beneficial associations with the prevention of obesity and other diseases [58].
Our results revealed that weekday nap duration (≤30min/d) was inversely related to WHR, which was consistent with previous studies including in Southwest China [52], Iran [59] and among Latino population [60]. Several hypotheses may be proposed. First, daytime nap can stimulate the sympathetic nervous system, which is positively with obesity [61,62]. But the specific mechanism of action seems to be unclear. Second, long nap duration may cause E-Type and more sleep latency even insomnia at night, which are obviously relevant with obesity.
Media use such as mobile phone, computer or television, etc. and increasing screen time before sleep or even after lights out have become a common practice specially among college students. In the current study, nearly 90% of the participants had screen time before sleep more than 0.5h, an astonishing proportion. Our results indicated that screen time before sleep (>0.5h) led to more potential to general and abdominal obesity (higher VFA, BFP, WHR, WHtR, etc.), which were consistent with previous findings. A study on Indian adolescents exposed excess screen time could affect adiposity indicators [63]. Although the exact underlying mechanisms remain unclear, we hypothesize that various amounts of blue light emitted by electronic screens could suppress melatonin production [64] and might be deleterious to sleep cycles. Studies regarding adolescentshave demonstrated that screen time during bedtime has impacts on poor sleep outcomes (shortersleep duration, longer sleep latency, lower sleep efficiency, etc.) [65,66], which can influence biological and social rhythms (e.g. evening chronotype, SJL) or circadian rhythm [74,67]. Interfering with excess screen time in bed and avoid sleep disorders, is a practical step to prevent obesity among adolescents effectively.
Several strengths of our study deserve mentioning. The use of InBody 720 captures non-invasive and highly precise assessments of fat distribution and body composition, during short measurement time. In addition, the selected indicators were representative enough to describe both general and abdominal obesity. Third, not only did we concentrate on different dimensions of sleep characteristics, but also we attached great importance to the weekday-to-weekend sleep differences in this special population such as sleep midpoint, SJL, etc. With the increasing popularity of electronic devices among young individuals, we focused on screen time before sleep to analyze the association with general and abdominal adiposity. Additionally, all descriptive analyses were stratified by gender and accordingly we controlled confounders including sex in linear regression analyses. We controlled for various kinds of potential confounding factors, including demographic variables, dietary intake, alcohol, smoking and physical exercise.
Some limitations of this study should be highlighted. First, owing to the cross-sectional design of our analysis, causal relationships could not be allowed. The participants were selected from one university, which made selection bias unavoidable. Thus, futural investigations should contain well-designed longitudinal studies and cover populations from different regions. Besides, instead of using objective measurements of sleep characteristics such as accelerometers, we applied subjective questionnaires. But it seemed that our approach was good enough to achieve our purpose among participants. What’s more, psychological factors such as depression and mood disturbances probably are related to both obesity and sleep. Thus, ongoing studies should include more influential factors to verify the results. Moreover, standing position was used in the BIA (InBody 720), which might bring a few inaccuracies of the measured parameters. To prefer the measurement of body composition in the future study, before contacting with the electrodes, participants should be guided to clean their hands and feet with antibacterial tissues provided by the manufacturer. Finally, we didn’t consider genetic markers of obesity, possibility of familial obesity inheritance of participants, and factors or conditions related to sleep disturbance, including sleep apnea, insomnia, etc.”
The following reference was supplemented in the Reference List (marked in blue):
42.Fan Y, Zhang L, Wang Y, Li C, Zhang B, He J, Guo P, Qi X, Zhang M, Guo C et al: Gender differences in the association between sleep duration and body mass index, percentage of body fat and visceral fat area among chinese adults: a cross-sectional study. BMC Endocr Disord 2021, 21(1):247.
58.Santos DCD, Oliveira Filho JG, Sousa TL, Ribeiro CB, Egea MB: Ameliorating effects of. metabolic syndrome with the consumption of rich-bioactive compounds fruits from Brazilian Cerrado: a narrative review. Crit Rev Food Sci Nutr 2022, 62(27):7632-7649.
References
- Obesity: preventing and managing the global epidemic. Report of a WHO consultation. World Health Organ Tech Rep Ser, 894, i-xii, 1-253.
2022a. Older adolescent (15 to 19 years) and young adult (20 to 24 years) mortality.
2022b. Older children and young adolescent mortality (5 to 14 years). [Online]. Available: https://www.who.int/news-room/fact-sheets/detail/older-children-and-young-adolescent-mortality-(5-to-14-years) [Accessed].
ADAN, A., ARCHER, S. N., HIDALGO, M. P., DI MILIA, L., NATALE, V. & RANDLER, C. 2012. Circadian typology: a comprehensive review. Chronobiol Int, 29, 1153-75.
AZZOPARDI, P. S., SAWYER, S. M., CARLIN, J. B., DEGENHARDT, L., BROWN, N., BROWN, A. D. & PATTON, G. C. 2018. Health and wellbeing of Indigenous adolescents in Australia: a systematic synthesis of population data. Lancet, 391, 766-782.
CATAROZOLI, C. 2023. Sleep During the Pandemic. Sleep Med Clin, 18, 219-224.
CHOI, G., YOON, H. J., SONG, Y. J., JEONG, H. M., GU, J. E., HAN, M., KIM, S. H., YOON, J. W. & KIM, H. 2022. Consistency of the estimated target weights and ECW/TBW using BIA after hemodialysis in patients between standing and lying-down positions. BMC Nephrol, 23, 106.
DOS SANTOS, R. R. G., FORTE, G. C., MUNDSTOCK, E., AMARAL, M. A., DA SILVEIRA, C. G., AMANTEA, F. C., VARIANI, J. F., BOOIJ, L. & MATTIELLO, R. 2020. Body composition parameters can better predict body size dissatisfaction than body mass index in children and adolescents. Eat Weight Disord, 25, 1197-1203.
GAO, M., WEI, Y. X., LYU, J., YU, C. Q., GUO, Y., BIAN, Z., PEI, P., DU, H. D., CHEN, J. S., CHEN, Z. M., HUANG, T., LI, L. M. & CHINA KADOORIE BIOBANK COLLABORATIVE, G. 2019. [The cut-off points of body mass index and waist circumference for predicting metabolic risk factors in Chinese adults]. Zhonghua Liu Xing Bing Xue Za Zhi, 40, 1533-1540.
JAHRAMI, H. A., ALHAJ, O. A., HUMOOD, A. M., ALENEZI, A. F., FEKIH-ROMDHANE, F., ALRASHEED, M. M., SAIF, Z. Q., BRAGAZZI, N. L., PANDI-PERUMAL, S. R., BAHAMMAM, A. S. & VITIELLO, M. V. 2022. Sleep disturbances during the COVID-19 pandemic: A systematic review, meta-analysis, and meta-regression. Sleep Med Rev, 62, 101591.
JANKOVIC, N., SCHMITTING, S., KRUGER, B., NOTHLINGS, U., BUYKEN, A. & ALEXY, U. 2021. Changes in chronotype and social jetlag during adolescence and their association with concurrent changes in BMI-SDS and body composition, in the DONALD Study. Eur J Clin Nutr.
LI, J. C., LYU, J., GAO, M., YU, C. Q., GUO, Y., BIAN, Z., PEI, P., DU, H. D., CHEN, J. S., CHEN, Z. M., LI, L. M. & CHINA KADOORIE BIOBANK COLLABORATIVE, G. 2019. [Association of body mass index and waist circumference with major chronic diseases in Chinese adults]. Zhonghua Liu Xing Bing Xue Za Zhi, 40, 1541-1547.
MCLESTER, C. N., NICKERSON, B. S., KLISZCZEWICZ, B. M. & MCLESTER, J. R. 2020. Reliability and Agreement of Various InBody Body Composition Analyzers as Compared to Dual-Energy X-Ray Absorptiometry in Healthy Men and Women. J Clin Densitom, 23, 443-450.
MONGRAIN, V., LAVOIE, S., SELMAOUI, B., PAQUET, J. & DUMONT, M. 2004. Phase relationships between sleep-wake cycle and underlying circadian rhythms in Morningness-Eveningness. J Biol Rhythms, 19, 248-57.
MONTARULI, A., CASTELLI, L., MULE, A., SCURATI, R., ESPOSITO, F., GALASSO, L. & ROVEDA, E. 2021. Biological Rhythm and Chronotype: New Perspectives in Health. Biomolecules, 11.
MURILLO-CASTILLO, K. D., FRONGILLO, E. A., CORELLA-MADUENO, M. A. & QUIZAN-PLATA, T. 2020. Food Insecurity Was Associated with Lower Fruits and Vegetables Consumption but Not with Overweight and Obesity in Children from Mexican Fishing Communities. Ecol Food Nutr, 59, 420-435.
NIE, X., CHEN, J., MA, X., NI, Y., SHEN, Y., YU, H., PANAGIOTOU, G. & BAO, Y. 2020. A metagenome-wide association study of gut microbiome and visceral fat accumulation. Comput Struct Biotechnol J,18, 2596-2609.
PAN, X. F., WANG, L. & PAN, A. 2021. Epidemiology and determinants of obesity in China. Lancet Diabetes Endocrinol, 9, 373-392.
PFEFFERBAUM, A., KWON, D., BRUMBACK, T., THOMPSON, W. K., CUMMINS, K., TAPERT, S. F., BROWN, S. A., COLRAIN, I. M., BAKER, F. C., PROUTY, D., DE BELLIS, M. D., CLARK, D. B., NAGEL, B. J., CHU, W., PARK, S. H., POHL, K. M. & SULLIVAN, E. V. 2018. Altered Brain Developmental Trajectories in Adolescents After Initiating Drinking. Am J Psychiatry, 175, 370-380.
SANTOS, D. C. D., OLIVEIRA FILHO, J. G., SOUSA, T. L., RIBEIRO, C. B. & EGEA, M. B. 2022. Ameliorating effects of metabolic syndrome with the consumption of rich-bioactive compounds fruits from Brazilian Cerrado: a narrative review. Crit Rev Food Sci Nutr, 62, 7632-7649.
SHUAI, M., ZUO, L. S., MIAO, Z., GOU, W., XU, F., JIANG, Z., LING, C. W., FU, Y., XIONG, F., CHEN, Y. M. & ZHENG, J. S. 2021. Multi-omics analyses reveal relationships among dairy consumption, gut microbiota and cardiometabolic health. EBioMedicine, 66, 103284.
WIECH, P., WOLOSZYN, F., TROJNAR, P., SKORKA, M. & BAZALINSKI, D. 2022. Does Body Position Influence Bioelectrical Impedance? An Observational Pilot Study. Int J Environ Res Public Health, 19.
XIE, Y., WU, X., TAO, S., WAN, Y. & TAO, F. 2022. Development and validation of the self-rating of biological rhythm disorder for Chinese adolescents. Chronobiol Int, 39, 198-204.
ZERON-RUGERIO, M. F., CAMBRAS, T. & IZQUIERDO-PULIDO, M. 2019. Social Jet Lag Associates Negatively with the Adherence to the Mediterranean Diet and Body Mass Index among Young Adults. Nutrients, 11.
ZHAO, Q., SULLIVAN, E. V., HONNORAT, N., ADELI, E., PODHAJSKY, S., DE BELLIS, M. D., VOYVODIC, J., NOONER, K. B., BAKER, F. C., COLRAIN, I. M., TAPERT, S. F., BROWN, S. A., THOMPSON, W. K., NAGEL, B. J., CLARK, D. B., PFEFFERBAUM, A. & POHL, K. M. 2021. Association of Heavy Drinking With Deviant Fiber Tract Development in Frontal Brain Systems in Adolescents. JAMA Psychiatry, 78, 407-415.
